# Atlasing white matter and grey matter joint contributions to resting-state networks in the human brain

Victor Nozais [1,2✉], Stephanie J. Forkel[2,3,4,5], Laurent Petit [1], Lia Talozzi [2,6], Maurizio Corbetta [7], Michel Thiebaut de Schotten [1,2] & Marc Joliot [1✉]

Over the past two decades, the study of resting-state functional magnetic resonance imaging has revealed that functional connectivity within and between networks is linked to cognitive states and pathologies. However, the white matter connections supporting this connectivity remain only partially described. We developed a method to jointly map the white and grey matter contributing to each resting-state network (RSN). Using the Human Connectome Project, we generated an atlas of 30 RSNs. The method also highlighted the overlap between networks, which revealed that most of the brain's white matter (89%) is shared between multiple RSNs, with 16% shared by at least 7 RSNs. These overlaps, especially the existence of regions shared by numerous networks, suggest that white matter lesions in these areas might strongly impact the communication within networks. We provide an atlas and an open-source software to explore the joint contribution of white and grey matter to RSNs and facilitate the study of the impact of white matter damage to these networks. In a first application of the software with clinical data, we were able to link stroke patients and impacted RSNs, showing that their symptoms aligned well with the estimated functions of the networks.

[1] Univ. Bordeaux, CNRS, CEA, IMN, UMR 5293, GIN, F-33000 Bordeaux, France. [2] Brain Connectivity and Behaviour Laboratory, Sorbonne Universities, Paris, France. [3] Donders Institute for Brain Cognition Behaviour, Radboud University, Nijmegen, the Netherlands. [4] Centre for Neuroimaging Sciences, Department of Neuroimaging, Institute of Psychiatry, Psychology and Neuroscience, King's College London, London, UK. [5] Departments of Neurosurgery, Technical University of Munich School of Medicine, Munich, Germany. [6] Department of Neurology, Stanford University, Stanford, CA, USA. [7] Department of Neuroscience, Venetian Institute of Molecular Medicine and Padova Neuroscience Center, University of Padua, Padova, PD 32122, Italy. ✉email: victor.nozais@gmail.com; marc.joliot@u-bordeaux.fr

Since the early 1990s, functional magnetic resonance imaging (fMRI) peers inside the workings of the living human brain[1]. Task fMRI unveiled countless aspects of brain functioning in healthy participants and patients. However, paradigm-free resting-state fMRI (rs-fMRI) analysis shows a striking correspondence with tasks-related fMRI[2] yet provides the most comprehensive depiction of the brain's functional organisation. Rs-fMRI explores the awake brain at rest when no specific external task is required from the participant. During rest, quasi-periodic low-frequency oscillations in the fMRI signal — blood-oxygen-level-dependent signal or BOLD — spontaneously occur[3]. Distant brain regions display synchronous BOLD signal oscillations, testifying to functional connectivity between regions and forming intrinsic functional networks, so-called resting-state networks (RSNs)[4–6]. RSNs are related to cognition[2], and their alteration has been linked to various brain pathologies[7–9], potentially opening up this field to a wide range of applications[10]. Hence, a resting-state acquisition is appealing and much less demanding than the active participant involvement in a task.

The identification of RSNs has been tackled in multiple ways[11]. One of the most popular approaches is an independent component analysis (ICA)[5,12,13], a data-driven method of signal separation[14] able to identify and extract independent components (ICs) corresponding to RSNs in the resting-state signal across the brain. From such components, resting-state networks and their grey matter maps can be identified.

With the progress of the functional connectivity framework, the question of the underlying structural connectivity became pressing. Indeed, understanding the anatomical drivers of the functional connection between multiple regions is necessary to properly study these networks' dynamics and biological relevance. In that regard, the advent of diffusion-weighted imaging (DWI) tractography enabled the description of white matter circuits in the living human brain. DWI measures the preferential orientations of water diffusion in the brain[15], which mostly follow axonal directions. Using orientation information, tractography algorithms piece together local estimates of water diffusion to reconstruct white matter pathways[16]. DWI is a potent, non-invasive in-vivo tool for mapping the white matter anatomy[17] and estimating structural connectivity between brain regions[18,19]. Leveraging tractography, the joint study of functional and structural connectivity has become an active field of research. However, previous work compared functional connectivity and structural connectivity between pairs of grey matter brain parcels[20,21]. Or when studies provided white matter maps related to resting-state networks, they either focused on a single network[22–25] or a restricted number of RSNs[26–28] with limited statistical confirmation of structural-functional connectivity relationships[22–25].

Notably, ICA applied to white matter tractography data produces circuits whose grey matter projections resemble resting-state networks[29,30]. These results demonstrate that information about the organisation of RSNs can also be extracted from white matter data and might be complementary to the information provided by resting-state BOLD signal analysis. However, to our knowledge, a comprehensive description of the white matter circuits in all identifiable resting-state networks is still lacking. In principle, such endeavour could be achieved by using the Functionnectome[30,31]. This recently developed method combines fMRI with tractography by projecting the grey matter BOLD signal onto white matter pathways.

In the present study, we extended our previous approach — the Functionnectome methodology[30] — to RSNs, integrating the grey matter resting-state signal with white matter connections, and analysed the resulting data through ICA. We produced the most comprehensive atlas of 30 RSNs specifying their grey matter maps together with their white matter circuitry — the WhiteRest atlas. This atlas unlocks the systematic exploration of white matter components supporting resting-state networks. The atlas comes with companion software, the WhiteRest tool, a module of the Functionnectome that will facilitate this exploration and assist the investigation of brain lesions' effects on RSNs and cognition.

## Results

**Mapping the resting brain: RSNs in white matter and grey matter.** Rs-fMRI scans derived from the Human Connectome Project[32] were converted into functionnectome volumes using the Functionnectome software[30,31] (available at http://www.bcblab.com). The original rs-fMRI and functionnectome volumes were simultaneously entered into an Independent Component Analysis for each participant. The resulting individual independent components were then automatically classified using MICCA[33], generating 30 IC groups, each group corresponding to one resting-state network. These groups were used to create RSN z-maps with paired white matter and grey matter maps (Fig. 1 and Fig. 2) — the WhiteRest atlas.

The paired white matter and grey matter z-maps generated by our method were thresholded using an arbitrarily high threshold of 7 to get a highly conservative estimate of the RSNs' spatial extent. Using this threshold, the combined white matter maps cover 96% of the brain white matter, except for some orbito-frontal and ventro-temporal pathways, part of the internal capsule and part of the brain stem. Similarly, the combined grey matter maps cover 79% of the cortical grey matter, except for ventral areas in the temporal and frontal lobes.

The WhiteRest atlas reveals both the functional grey matter of an RSN and this network's structural white matter circuitry. In the WhiteRest atlas, 21 of the 30 RSNs display a symmetrical pattern between the left and the right hemispheres. Nine networks are strongly lateralised, with four pairs of networks with contralateral homotopic counterparts, and one network that was exclusively left lateralised (RSN20, language production network). To help further explore each RSN, a description of the maps of all the RSNs can be found in the supplementary material (Supplementary Figs 1–30; the continuous maps are also available at https://identifiers.org/neurovault.collection:11895 for the white matter, and https://identifiers.org/neurovault.collection:11937 for the grey matter). As an illustrative example, the Default Mode Network (DMN) maps are showcased in Fig. 3. Although the DMN can be described as a set of sub-networks, one of them is most representative of what is usually called "DMN" in the literature[24,26,34]: the RSN18, which we labelled as "DMN proper".

The grey matter map of the DMN proper revealed the bilateral involvement of the medial frontal cortex (the medial superior frontal gyrus, the gyrus rectus, and the frontal pole), the superior frontal gyrus, the middle temporal gyrus, the precuneus, the angular gyrus and the cerebellum. The white matter maps of the RSN showed previously described pathways of the DMN, such as the second branch of the superior longitudinal fasciculus (SLF2) connecting the superior parietal lobe to the superior frontal gyrus and the cingulum connecting the precuneus area to the medial frontal area. Additionally, the middle temporal gyrus and the angular gyrus are connected by the posterior segment of the arcuate fasciculus. Interhemispheric connections were also present within the anterior and posterior corpus callosum connecting both frontal lobes and both precunei, respectively.

While the description of a known RSN, such as the DMN, can be used to validate the atlas, WhiteRest can also explore the uncharted white matter anatomy of RSNs, for instance, the Dorsal Attention Network (RSN13) presented in Fig. 4.

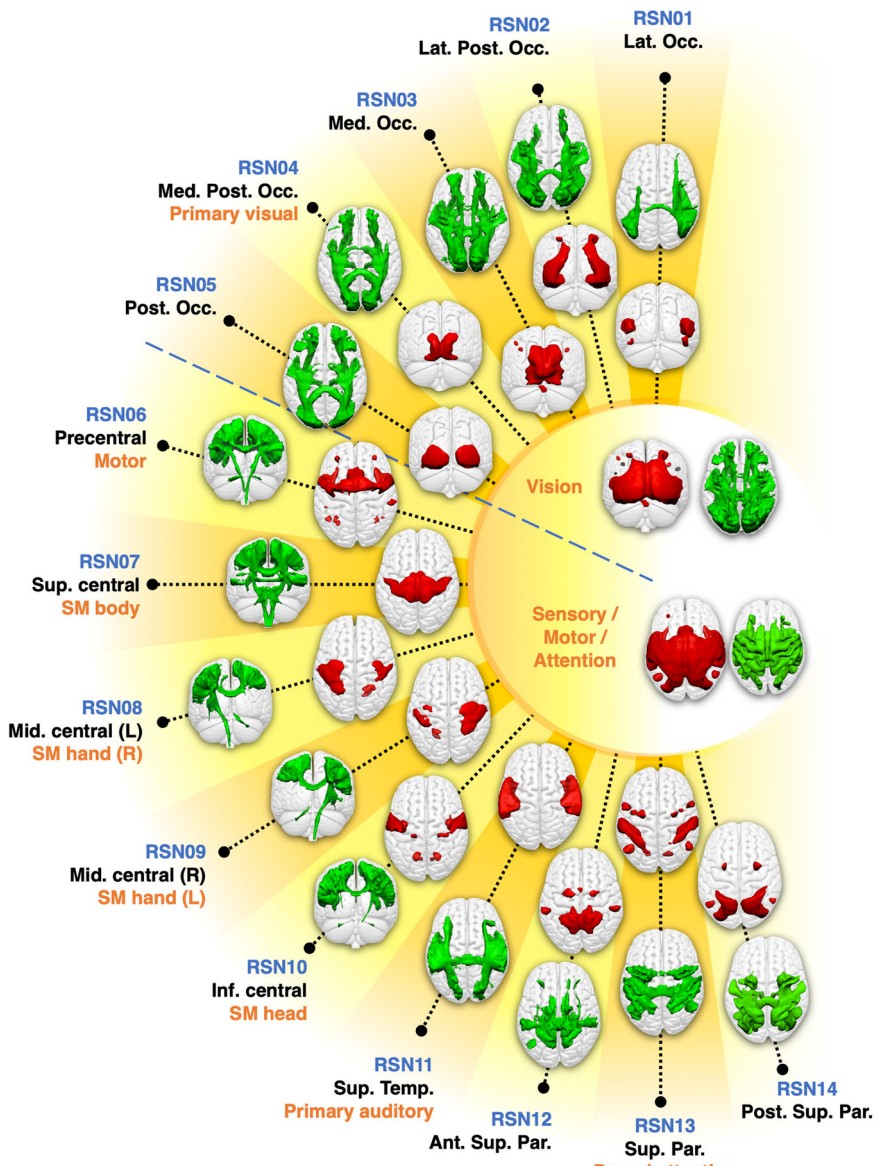

**Fig. 1 WhiteRest resting-state atlas of the visual and sensory/motor/attention domains.** This composite figure shows the white matter 3D maps (green) and grey matter 3D maps (red). Centre of the figure: Functional domains of the corresponding RSNs. The functional domains' 3D maps are the union of the associated RSNs. Labelling indicates an arbitrary RSN number (in blue), the primary cortical anatomical landmarks (in black) and putative cognitive function (in orange). Ant. Sup. Par.: Anterior superior parietal network; Inf. central — SM head: Inferior central network (somatomotor, head portion); Lat. Occ.: Lateral occipital network; Lat. Post. Occ.: Lateral posterior occipital network; Med. Occ.: Medial occipital network; Med. Post. Occ.: Medial posterior occipital network; Mid. central (L) — SM hand (R): Middle central network, left hemisphere component (somatomotor, right-hand portion); Mid. central (R) — SM hand (L): Middle central network, right hemisphere component (somatomotor, left hand portion); Post. Occ.: Posterior occipital network; Post. Sup. Par.: Posterior superior parietal network; Sup. central — SM body: Superior central network (somatomotor, body portion); Sup. Temp: Superior temporal network.

The grey matter map revealed the involvement of core regions of the DAN, with the parietal cortex – supramarginal gyrus (SMg), intraparietal sulcus (IPs) and superior parietal lobule (IPL) – and part of the superior frontal gyrus (SMg), in the frontal eye field region. It also showed other areas associated with the DAN, namely the precentral gyrus (PrCg), the insula and the posterior part of the middle temporal gyrus (MTg). The white matter map unveiled the involvement of the second branch of the superior longitudinal fasciculus (SLF2), connecting the inferior parietal cortex (IPs, SMg) with the frontal regions of the network (i.e. SFg, PrCg and insula). SFg and PrCg were also interconnected via the frontal aslant tract. The map also showed the involvement of the posterior segment of the arcuate fasciculus, connecting the MTg

with the parietal cortex. Additionally, the map revealed the involvement of the corpus callosum, ensuring interhemispheric connectivity.

**White matter RSNs, overlaps, and stroke lesions**. The WhiteRest atlas suggests that most RSNs share white matter pathways with other RSNs. Indeed, most of the brain's white matter (i.e. 89%) is shared amongst multiple RSN, with 16% of the white matter shared by at least 7 RSNs. By comparison, the grey matter contribution to RSNs show much less overlap, where 53% of the grey matter uniquely contributes to one RSN, and 45% to 2 or 3 RSNs. To determine the exact extent of the overlaps in the white matter, we generated an overlap map displaying the

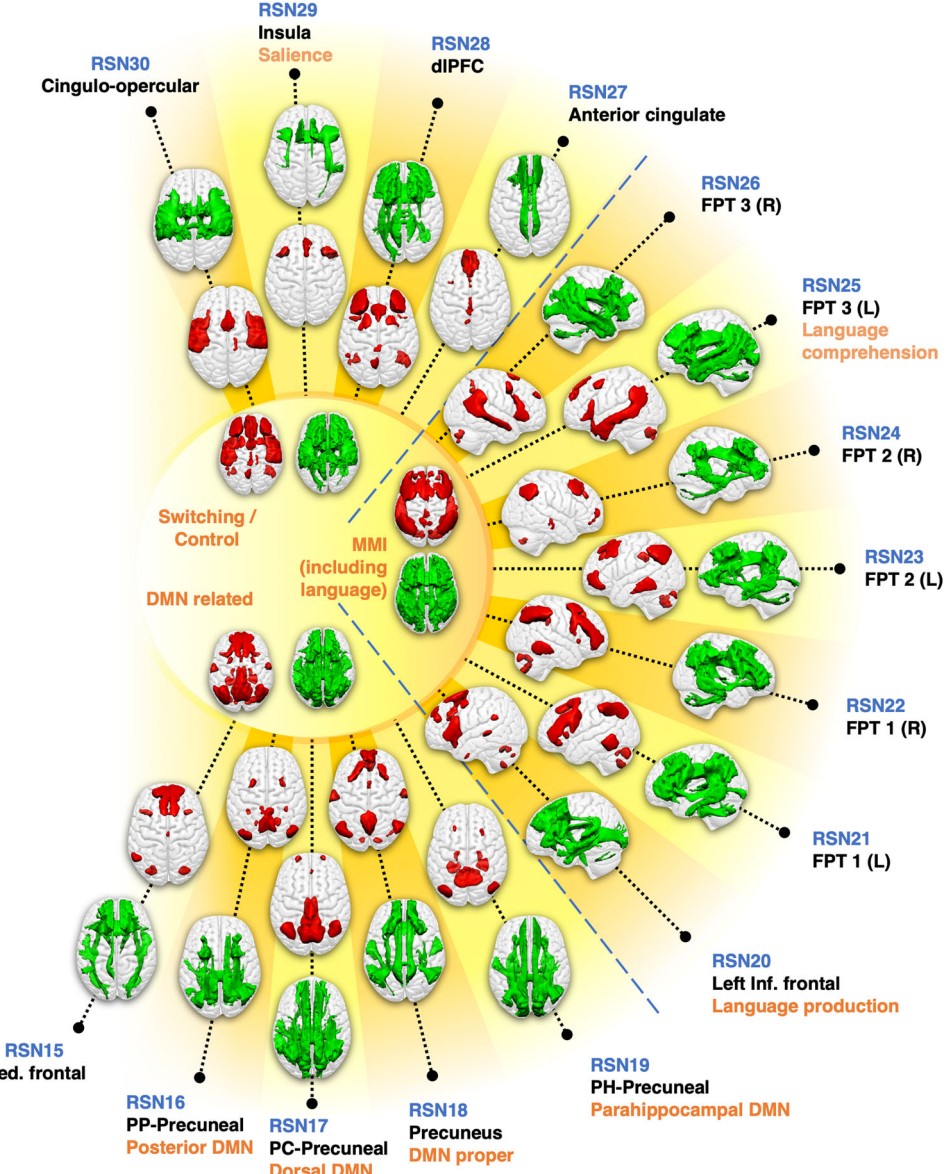

**Fig. 2 WhiteRest resting-state atlas of the switching/control, manipulation and maintenance of information (MMI), and default mode network (DMN) related domains.** This composite figure shows the white matter 3D maps (green) and grey matter 3D maps (red). Centre of the figure: Functional domains of the corresponding RSNs. Labelling indicates an arbitrary RSN number (in blue), the primary cortical anatomical landmarks (in black) and putative cognitive function (in orange). DMN: Default Mode Network. dlPFC: Dorso-lateral prefrontal cortex network; FPT 1/2/3 (L/R): Fronto-parieto-temporal network 1/2/3, Left/Right hemisphere component; Med. frontal: Medial frontal network; PC-Precuneal: Posterior cingulate-precuneal network; PH-Precuneal: Parahippocampal-Precuneal network; PP-Precuneal: Posterior parietal-precuneal network.

number of RSN per voxel in the brain. Large areas of the deep white matter showed high RSN overlap count (>7 overlapping RSNs), including in the centrum semiovale and sub-portions of the medial corpus callosum (Fig. 5). RSNs also overlapped highly in the cingulum, the second and third branches of the superior longitudinal fasciculi (SLF2, SLF3), the arcuate fasciculi, and the inferior fronto-occipital fasciculi (IFOF) in both hemispheres. In contrast, the superficial white matter demonstrated less RSN overlap.

The existence of areas with high-density RSN overlap in the white matter point toward the idea that lesions to the white matter could severely impact the functioning of multiple RSNs and hence cause a diverse pattern of clinical symptoms. To explore this aspect, we developed a new module, the WhiteRest tool, freely available online through the Functionnectome software (available at http://www.bcblab.com). The WhiteRest tool

estimates the white matter disruption of an RSN by a lesion with a "disconnectome-RSN overlap" score, the DiscROver score. It can also measure the local involvement of each RSN for any given region of interest (ROI) in the white matter (measured as "Presence score", see the WhiteRest user guide in the supplementary material).

We validated the WhiteRest atlas in a clinical dataset of 131 stroke patients[35] and compared their neurobehavioral deficits with the measured impact of the lesion on the RSNs. More specifically, we explored the three deficits which could clearly be associated with RSNs based on their estimated function. The three deficits and the four RSNs in question were: left upper-limb motor control (MotorL) deficit associated with the somatomotor network of the left-hand (RSN 09) (Fig. 6a); right upper-limb motor control (MotorR) deficit associated with the somatomotor network of the right-hand (RSN 08) (Fig. 6b); and language

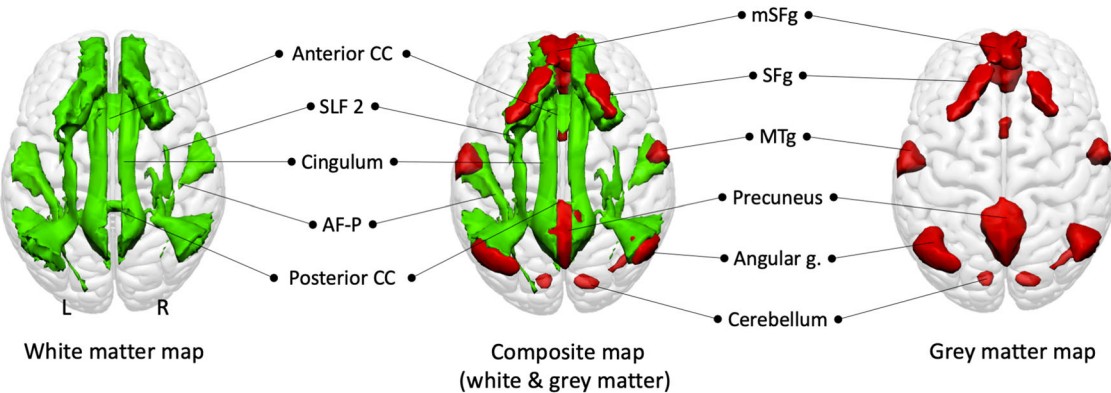

**Fig. 3 Default Mode Network proper (RSN18) maps, dorsal view.** White matter map in green, grey matter map in red. Composite map in the middle. The cerebellum is visible through the glass-brain effect. AF-P: Arcuate fasciculus (posterior segment); mSFg.: medial superior frontal gyrus; MTg: Middle temporal gyrus; SFg: Superior frontal gyrus; SLF2: Second branch of the superior longitudinal fasciculus.

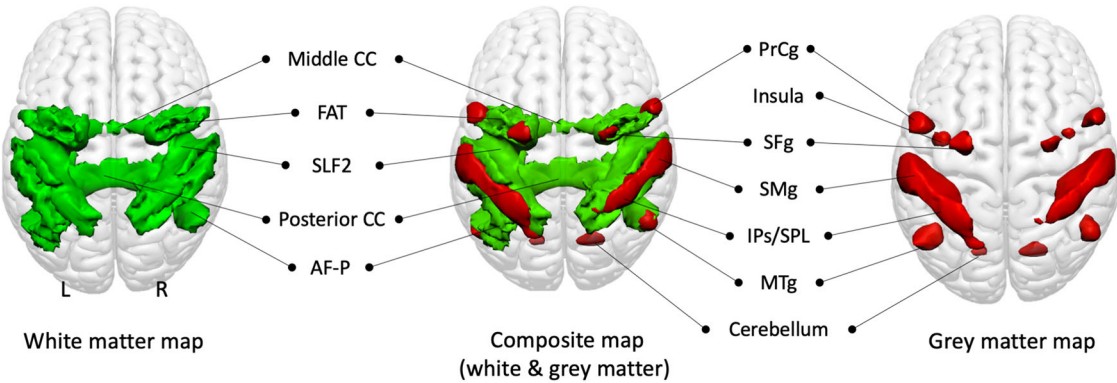

**Fig. 4 Dorsal Attention Network (RSN13) maps, dorsal view.** White matter map in green, grey matter map in red. Union of the two maps in the middle. The insula and cerebellum are visible through the glass-brain effect. AF-P Arcuate fasciculus (posterior segment), FAT Frontal Aslant Tract, IPs/SPL Intraparietal sulcus and superior parietal lobule, Middle CC Middle part of the corpus callosum, MTg Middle temporal gyrus, Posterior CC Posterior part of the corpus callosum, PrCg Precentral gyrus, Precentral s. Precentral sulcus, SFg Superior frontal gyrus, SLF2 Second branch of the Superior Longitudinal Fasciculus, SMg: Supramarginal gyrus.

deficit associated with the language comprehension network (RSN 25) (Fig. 6c) and with the language production network (RSN 20)(Fig. 6d). Each deficit score (MotorL, MotorR, and Language deficit) is derived from a principal component analysis (PCA) of the set of neurobehavioral assessment scores related to the deficit. The impact of a lesion on RSNs was measured with the DiscROver score from the WhiteRest tool. We show a strong and highly significant correlation between the neurobehavioral deficit scores and the DiscROver scores for the related RSNs. The Pearson correlation between the scores was: 0.75 ($R^2 = 0.57$, $n = 131$) for the "Left upper-limb"; 0.60 ($R^2 = 0.36$, $n = 131$) for the "Right upper-limb"; 0.68 ($R^2 = 0.46$, $n = 131$) for the "Language (comprehension)"; and 0.61 ($R^2 = 0.37$, $n = 131$) for the "Language (production)". All correlations were highly significant with $p < 10^{-13}$. For a more qualitative overview, we showed that the lesion of all patients with strong deficits (deficit score in the upper decile, Supplementary Fig. 31) overlapped with the studied RSN (Supplementary Fig. 32–38). For comparison, we also provided the illustrations of lesions and RSNs for mild deficit (deficit score between the 60% quantile and the 75% quantile), showing that the overlaps were then much more limited (Supplementary Fig. 39–42).

Using this dataset, we also tested the plausibility of our above-mentioned hypothesis whether lesions impacting multiple RSNs would "cause a diverse pattern of clinical symptoms". To do so, we selected patients for whom at least a third (DiscROver score >

33) of both the right-hand somatomotor RSN and the language comprehension RSN were impacted. Among these few patients ($n = 11$), the majority ($n = 9$) had clear symptoms (i.e., deficit score in the upper quartile) for both language and right upper-limb motor control (Supplementary Fig. 43 & 44). While the group size of this analysis is too small for definitive conclusions and limited to two RSNs, we believe these preliminary results are encouraging and ought to incentivize more research into this issue. It should however be noted that they are not enough to validate direct clinical applications of the atlas, but do show promise for its potential use as a tool in clinical research.

## Discussion

We introduce WhiteRest, an atlas derived from integrated functional signal and structural information revealing white matter and grey matter components for each resting-state network. As such, the present work showcases two original results. First is the atlas, which consists of the systematic mapping of white matter that contributes to the resting-state networks. Second, our results demonstrate that white matter pathways can contribute to multiple RSNs. This new atlas offers the prospect of exploring the impact of white matter lesions on the integrity of resting-state networks and, thus, their functioning.

The WhiteRest atlas is, to our knowledge, the first comprehensive statistical mapping of the white matter contribution to RSNs. We generated white and grey matter maps concurrently,

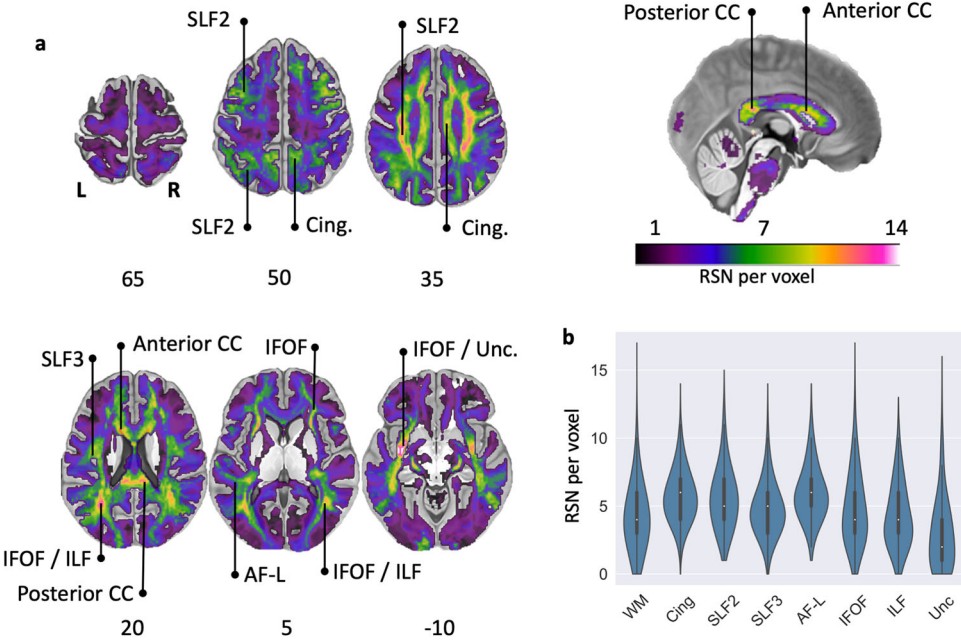

**Fig. 5 RSN overlap in the brain. a** Overlay map of RSN white matter maps. Colour bar: Number of RSN per voxel (saturated for *n* > 14). Anterior CC Anterior corpus callosum, AF-L Arcuate fasciculus (long segment), Cing. Cingulum, IFOF Inferior fronto-occipital fasciculus, ILF Inferior longitudinal fasciculus, Posterior CC Posterior corpus callosum, SLF2 Second branch of the superior longitudinal fasciculus, SLF3 Third branch of the superior longitudinal fasciculus, Unc Uncinate fasciculus. **b** Violin plots (normalised by plotted area) of the overlap values in the total white matter and along the studied pathways (left and right hemispheres combined). Each plot also contains a boxplot with the median, the interquartile range (IQR), and "whiskers" extending within 1.5 IQRs of the lower and upper quartile. WM: average whole white matter.

yielding continuous statistical maps of the RSNs in both tissues, thus, allowing for a thorough exploration of each network. The combination of functional and structural information can help the exhaustive detection of RSNs as there is evidence that structural connectivity holds complementary information regarding RSNs[29]. Hence, the multimodality of the signal might help identify and segregate networks as previously demonstrated by other groups with different modalities (e.g., Glasser's multi-modal parcellation[36]). Previous studies also combined grey matter functional and white matter structural information to explore the white matter contribution to resting-states networks but were limited to a low number of RSNs[25,27], or were focused on the white matter support of dynamical changes in functional connectivity[37]. In contrast, recent works that undertook the atlasing of the RSN white matter connectivity did not directly combine functional and structural information. They mapped the RSN white matter circuits by connecting RSNs cortical regions from a pre-existing cortical RSNs atlas, using tractography data[28,38]. In this approach, the functional-structural mapping is highly dependent on the original cortical RSN atlas, while in our method, grey and white matter information are used concurrently.

Another intriguing approach to the functional study of white matter has recently been gaining traction and shown auspicious results: the analysis of the BOLD signal directly in the white matter (mini-review by Gore et al., 2019). Using the BOLD signal from white matter allows for its functional exploration and mapping without resorting to connectivity models, which may lead to more physiologically accurate descriptions. Multiple studies have used this framework to unveil RSNs in white matter, successfully adapting classical RSN investigation methods to the white matter[39–42]. These studies revealed a functional parcellation of the white matter, showing that it was possible to identify multiple RSNs purely from functional signals while staying consistent with the underlying structural connectivity. However,

these approaches have yet to produce a functional parcellation of the white matter displaying continuous, long-range connectivity between different cortical regions. While efforts have been made to link white matter RSNs with grey matter RSNs, previous studies were unable to present a consistent 1-to-1 correspondence between white and grey matter RSNs. As our objective is to investigate the white matter connectivity underlying traditional grey matter RSNs, the analyses directly using white matter BOLD signals do not appear to offer immediately interpretable results on this matter. In contrast, by combining structural and functional (grey matter) signals with the Functionnectome, our approach generated white matter maps that could better represent each network, and systematically paired them with their well-known grey matter counterparts. The WhiteRest atlas also demonstrated overlaps between RSNs, consistent with fibres from distinct networks crossing in the white matter. Nevertheless, combining both approaches in the future (white matter BOLD analysis and Functionnectome) could be highly beneficial as it could allow for a finer understanding of the functional involvement of white matter in resting-state.

Our data-driven method allowed for a global approach by mapping the whole brain, except for ventral areas in zones strongly affected by magnetic susceptibility artefacts, where both the fMRI and diffusion signals are degraded[43]. The individual-ICA-based scheme used to produce the statistical group maps revealed a fine granularity of the RSNs, where brain regions that are spatially distant but functionally and structurally connected are attributed to the same RSN. The fine granularity of the default mode network (DMN) in the WhiteRest atlas is a good example of the multimodal improvement of the networks' segregation. Our analysis replicated four previously described[44] DMN-related RSNs involving the precuneus (RSN 16, 17, 18 & 19), while also differentiating a DMN proper (RSN18) from a medial frontal network (RSN15). For the DMN proper, the structural connectivity is largely known[22–28,45], which offers a good opportunity to validate

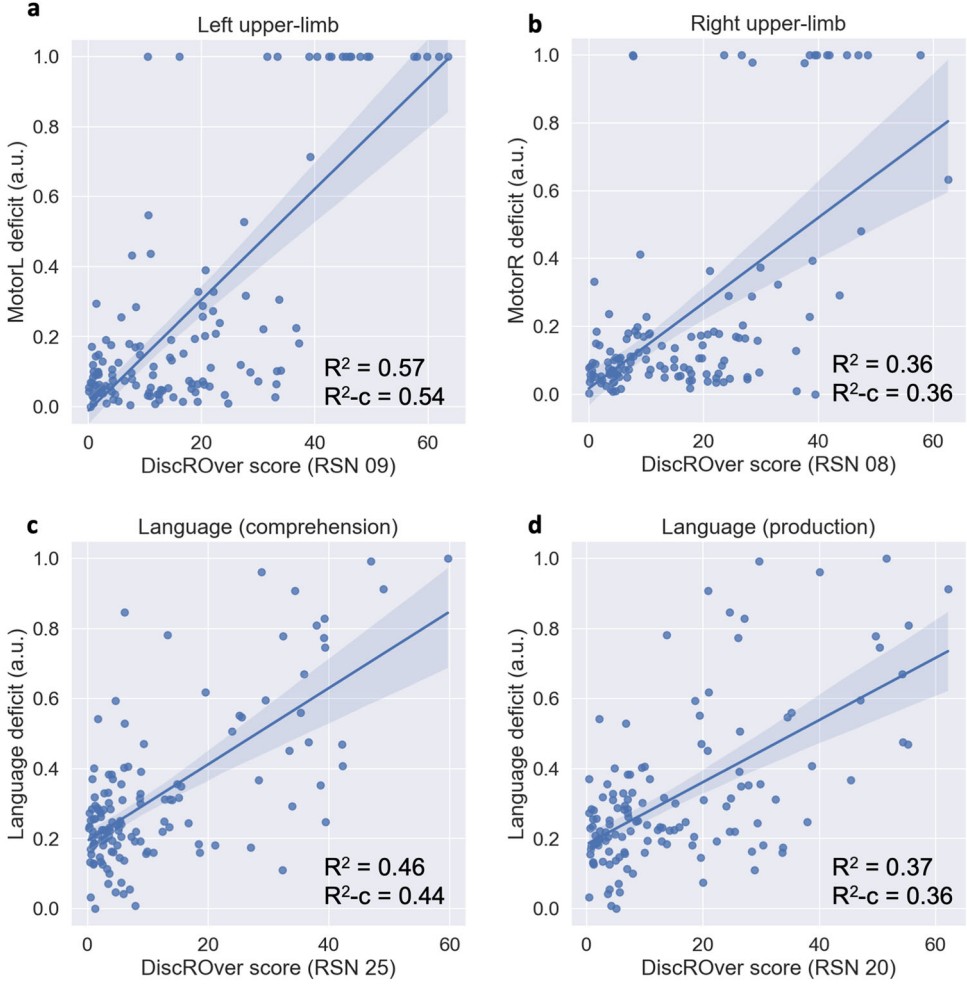

**Fig. 6 Relationship between neurobehavioral deficit and WhiteRest DiscROver. a, b** Left (**a**) and right (**b**) upper-limb motor control deficit vs. DiscROver score for the Somatomotor network of the left (**a**) and right (**b**) hand; **c, d** Language deficit vs. DiscROver score for the language comprehension network (**c**) and the language production network (**d**). In each graph, all the patients are represented (n = 131) and the blue line corresponds to the linear fit between the scores, and the light blue area corresponds to the confidence interval (set at 95%) for the linear fit. a.u.: arbitrary unit (scores set between 0 and 1); $R^2$: Coefficient of determination. $R^2$-c : Coefficient of determination corrected by controlling for age, sex and chronicity.

our method. For instance, WhiteRest's DMN proper white matter map confirmed the involvement of the cingulum, connecting the precuneus with the frontal cortex[22–28,45], and of the superior longitudinal fasciculus (SLF2) connecting the superior frontal gyrus with the angular gyrus[24,25,27]. Similarly, the posterior segment of the arcuate fasciculus that connects the inferior parietal lobule with the posterior temporal lobe has also been reported in previous studies for the DMN[22,24]. Complementing the DMN-proper, following previous DMN descriptions[24], the medial frontal network involved the inferior longitudinal fasciculus (ILF, connecting the occipital lobe with the temporal lobe), the uncinate (connecting the temporal pole to the inferior frontal lobe), and the cingulum (connecting temporal-parietal-frontal areas).

Similarly, the WhiteRest atlas can be used in a prospective and explorative manner, as shown with the unveiling of the dorsal attention network (RSN13). While the grey matter architecture of the DAN is well documented[46,47], its white matter support has only been partially explored[48]. To our knowledge, WhiteRest reveals the first comprehensive description of the DAN's white matter that includes bilateral association fibres connecting ipsilateral regions, and commissural fibres ensuring interhemispheric connectivity. However, disentangling the exact functional relevance of each connection remains a challenge that will require,

for example, functionnectome investigation[30] or advanced lesions analyses[49–51]. Such approaches might shed light on the hierarchical and functional implications of RSN circuits[49,50,52,53]. Recent results have highlighted the importance of white matter structural disconnections in the disruption of functional connectivity[53], and this disruption has been linked to behavioural and cognitive dysfunction[54,55]. Therefore, being able to identify these RSN white matter "highways" would propel our understanding of disconnection symptoms, improve recovery prognostics, and inform preoperative brain surgery planning[56]. To facilitate these efforts, we released the WhiteRest tool (as a module of the Functionnectome) that quantifies the presence of RSNs in a specific region of the brain's white matter. The WhiteRest module was designed to accept regions of interest (e.g. from parcellations or lesions) in the MNI 152 space $(2 \times 2 \times 2 \text{ mm}^3)$ and estimates the RSNs involved or in the case of lesions, which RSNs would be impacted by a lesion in this region.

As a proof of concept and to validate the atlas, the WhiteRest tool was applied to the lesions of 131 stroke patients to compare the DiscROver score of 4 RSNs with the symptoms associated with their putative functions. We observed a strong correlation between each neurobehavioral deficit and their corresponding RSN DiscROver score, namely: Left and right upper-limb motor

control deficit with the somatomotor networks of left- and right-hand, respectively; and language deficit with both language production and language comprehension networks.

These results serve as the first clinical validation of the WhiteRest atlas, showing that its structural and functional mapping is sound, and that it could be employed in the scope of patient research, opening up a novel strategy to assert the cognitive functions related to RSNs. Associating functions to RSNs is usually done by indirect inference, using their spatial maps and contrasting them with fMRI-derived activation maps of specific cognitive functions[2]. As lesion studies have historically been a major tool in determining functions of grey matter area[57], and more recently of white matter pathways[58], WhiteRest provides a new tool to understand the link between cognition and resting-state networks. Reversely, the WhiteRest integrated functional and structural connectivity can shed light on the functional mechanisms of the brain and the origins of cognitive disorders. While promising results link stroke symptoms and RSNs in our study, further investigations will be required to fully disentangle the relationship between cognition (or cognitive deficits) and RSNs, using more advanced models than the relatively simple linear approach from the present study. Recent works have been undertaking the prediction of symptoms and recovery from stroke based on functional and structural data[58,59], a very important and interesting goal for which WhiteRest may eventually be of use, adding interpretable data to these multimodal methods.

While the WhiteRest module and atlas represent an advance in resting state functional neuroimaging, it is not exempt from limitations. For instance, we excluded the cerebellum-centred RSN in the present work. This decision was motivated by some limitations of tractography that are exacerbated in the cerebellum[60], mitigating the quality of the modelled pathways. For example, the fine structure of the cerebellum and the gathering of fibres in the brainstem are affected by partial volume and bottleneck effects[61]. Also, some of the maps displayed white matter pathways leading to grey matter areas absent on the related grey matter map. Some of these cases can be explained as simply threshold-dependent (i.e. $z > 7$ to facilitate the visualisation of 3D structures), which hid some of the less significant (but still involved) areas. However, these pathways might correspond to the structural link between different RSNs. Thus, when exploring a network in detail, we strongly advise checking the non-thresholded maps to better appreciate the entire white matter network involved in RSNs.

All in all, we introduced a novel combined atlas of resting-state networks based on functional and structural connectivity to deliver white matter and grey matter maps for each RSN — the WhiteRest atlas. This atlas allows for the exploration of the structural support of individual RSN and facilitates the study of the impact of white matter lesions on resting-state networks. Accordingly, we released the WhiteRest module that estimates the proportion of RSNs impacted by a given white matter lesion. With this tool, future research can focus on exploring the link between white matter lesions and their effects on the related resting-state networks in light of symptom diagnosis. Leveraging a deep-learning approach recently introduced[44] opens the possibility for individual resting-state functionnectome analyses and will facilitate a more personalised neuromedicine.

## Methods

**HCP dataset.** The dataset used in the present study is composed of the openly-available resting-state scans (rsfMRI) from 150 participants (75 females; age range 22-35 years) of the Human Connectome Project (HCP)[32], with 45 participants from the test-rest HCP dataset and 105 randomly sampled participants from the Young adult dataset (http://www.humanconnectome.org/study/hcp-young-adult/;

WU-Minn Consortium; Principal investigators: David Van Essen and Kamil Ugurbil; 1U54MH091657).

Full description of the acquisition parameters can be found on the HCP website (https://www.humanconnectome.org/hcp-protocols-ya-3t-imaging) and in the original HCP publication[62]. Briefly, the resting-state scans were acquired with 3 Tesla Siemens Skyra scanners and consist of whole-brain gradient-echo EPI acquisitions using a 32-channel head coil with a multiband acceleration factor of 8. The parameters were set with: TR = 720 ms, TE = 33.1 ms, 72 slices, 2.0 mm isotropic voxels, in-plane FOV = 208 × 180 mm, flip angle = 52°, BW = 2290 Hz/Px. Each resting-state acquisition consisted of 1200 frames (14 min and 24 sec), and was repeated twice using a right-to-left and a left-to-right phase encoding.

The resting-state acquisitions were then preprocessed using the "Minimal preprocessing pipeline" *fMRIVolume*[63], applying movement and distortion corrections and registration to the MNI152 (2009) nonlinear asymmetric space. Note that all the analyses done in the present study were conducted in this space, and subsequent mention of "MNI152" will refer to that space. Further processing steps were also applied: despiking; voxelwise detrending of potentially unwanted signal (6 motion regressors, 18 motion-derived regressors[64], and CSF, white matter, and grey matter mean time-courses); temporal filtering (0.01–0.1 Hz); and spatial smoothing (5 mm FWHM). While of exceptional quality, we chose to alter the HCP data to make it clinically relevant. A composite resting-state 4D volume was generated by discarding the 300 first and 300 last frames of the resting state acquisitions and concatenating (along the time axis) the resulting volumes. For each participant, this corresponded to 7.5 min with the left-right and 7.5 min with the right-left phase of acquisition (=1200 frames total).

**Stroke dataset.** A dataset of 131 stroke patients (46% female, 54 ± 11 years, range 19-83 years) with diverse cognitive deficits was used to validate the plausibility of the atlas and demonstrate the feasibility for potential clinically oriented approaches. The cohort of patients (n = 132) was recruited at the School of Medicine of Washington University in St. Louis (WashU)[35]. One patient from this cohort was excluded because of missing data. All participants gave informed consent, as per the procedure implemented by WashU Institutional Review Board and in agreement with the Declaration of Helsinki (2013). The data of each patient consisted of their MRI-derived manually segmented brain lesion as well as the associated neurobehavioral scores. In the present study, we focused on 3 deficits: language, left upper-limb motor control, and right upper-limb motor control deficits. They were established based on the acute (13 ± 4.9 days after stroke) neurobehavioral assessment scores of the patients, with 7 scores for language deficit, and 7 scores for each upper-limb motor control (left and right). The language deficit was tested using the Semantic (animal) verbal fluency test (SVFT, 1 score) and the Boston Diagnostic Aphasia Examinations (BDAE, 6 scores). The left and right upper-limb motor control deficit was tested using the Action Research Arm test (ARAT, 3 scores), the Jamar Dynamometer grip strength assessment (1 score), the 9-Hole Peg test (9HPT, 1 score), and the shoulder flexion and wrist extension assessment (2 scores). Additional precisions can be found in the supplementary methods.

**Extraction of white matter and grey matter components.** To explore the white matter structures of resting-state networks, we projected the functional signal from the rs-*f*MRI scans onto the white matter using the Functionnectome[30,31] (https://github.com/NotaCS/Functionnectome). The Functionnectome is a recently introduced method that unlocks the functional study of white matter. Briefly, the Functionnectome takes an *f*MRI scan and a grey matter mask as inputs combines grey matter BOLD signal with white matter anatomical priors, and outputs a new functional volume (called a functionnectome) with the same dimensions as the original *f*MRI scan (same 3D space, same number of time-points), but with the functional signal associated to the white matter. The Functionnectome provides default white matter priors[30]. The white matter priors were originally derived from the 7 Tesla diffusion data of a subset of 100 randomly selected HCP participants from the HCP young adults cohort. Deterministic tractography was run on this diffusion data using StarTrack (https://www.mr-startrack.com) to estimate the structural connectivity between each voxel of the brain and build the Functionnectome white matter priors.

In this functionnectome volume, the signal of a white matter voxel results from the combination of the BOLD signals from the voxels within the grey matter mask that are structurally connected to it (weighted by the probability of connection). The structural connectivity probability is given by the anatomical priors provided with the software (customisable priors option available). Using the Functionnectome thus allows the analysis of the functional signal in a connectivity-oriented framework by integrating the signal from distant but structurally connected grey matter voxels or clusters of voxels.

For our analysis, each of the 150 rs-*f*MRI scans from the dataset were processed with the Functionnectome, along with a grey matter mask (the same mask for all the subjects). This mask was computed using the brain parcellation data from all the participants: the mask corresponds to the voxels identified as part of the grey matter in at least 10% of the participants. This processing produced 150 resting-state functionnectome (rs-functionnectome) volumes, one per participant.

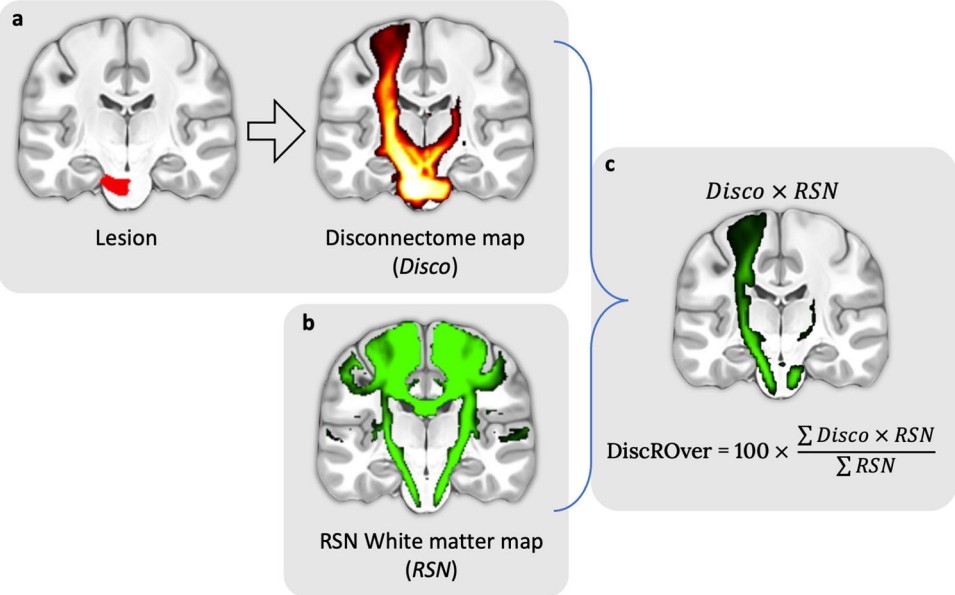

**Fig. 7 Steps for the computation of the DiscROver score. a** Lesion mask (left) and associated disconnectome (right). **b** RSN map used for the DiscROver score computation. **c** Visual representation of the weighted overlap, and computation of the DiscROver score. Disco: Disconnectome map; RSN: Resting-state network map.

To extract RSN from the data, we used an independent component analysis (ICA) method. For each participant, the original rs-*f*MRI scan was spatially concatenated with the associated rs-functionnectome. It resulted in functional volumes containing, side by side, the original resting-state signal (on the grey matter) and the rs-functionnectome signal (on the white matter). These composite functional volumes were then analysed with MELODIC (multivariate exploratory linear optimised decomposition into independent components, version 3.15), from the FMRIB Software Library (FSL)[65] to extract independent components (ICs) from the signal. The number of IC per participant was individually determined by Laplace approximation[66]. This resulted in a set of ICs, unlabeled putative RSNs, per participant. Each IC was composed of a temporal component (the IC's time-course) and a spatial map, displaying side by side (due to the above-mentioned spatial concatenation) the component in rs-*f*MRI (i.e. grey matter) space and in the rs-functionnectome (i.e. white matter) space. Each IC was then split into paired white matter maps and grey matter maps.

**Generating RSN maps by clustering ICs**. We used MICCA[33], an unsupervised classification algorithm developed to automatically group ICs from multiple individual ICAs (from different participants) based on the similarity of their spatial maps. The resulting groups, composed of ICs reproducible across participants, were used to produce group maps. Such an individual-based ICA scheme was preferred to the classical group ICA as some evidence suggests that group ICA can fail to separate some RSNs if they share functional hubs[33].

The atlas was produced by applying MICCA using the procedure described in the original Naveau et al. paper[33], in a 5-repetition scheme (i.e. ICA and MICCA were repeated 5 times per participant, and the resulting IC groups were paired with ICASSO[67]. The procedure generated 36 IC groups and their associated z-map, reproducible across the repetitions. Among them, 5 groups were identified as artefacts and were excluded, and 1 was located in the cerebellum and was excluded too in later analyses. The artefacts were visually identified when the grey matter z-map was spread along the border of the brain mask (typical of motion artefacts, $n = 3$), or was mainly located in ventricles or along blood vessels ($n = 2$). The cerebellar RSN was discarded because of known problems with tractography in the cerebellum[60], to avoid providing the atlas with a white matter map poorly representing the correct connectivity of this region.

We thus obtained a total of 30 RSNs, producing the WhiteRest atlas. Each RSN was then named by experts (MJ, VN) according to its anatomical localisation and in reference to AAL[68,69] and AICHA[68] atlas. Likewise, the classification of RSNs to a functional domain was done by an expert using the grey matter spatial patterns and estimated functional role of the RSNs presented here, compared to the one from Doucet et al.[12]

Note that we applied MICCA on the grey matter maps of the ICs. We used these maps for the clustering as MICCA has been developed and validated to cluster only classical resting-state derived spatial maps (in grey matter space). As each grey matter map is associated with a white matter map (since they are part of the same IC), the procedure still produces paired grey and white matter RSN maps, as presented in the atlas.

**Overlap analysis and DiscROver**. To measure the extent of overlaps between RSNs in the white matter, all the maps were thresholded (z > 7), binarized, and summed, generating a new map with the number of RSN per voxel.

Additionally, we provide a new software, the WhiteRest tool, to explore how the white matter is shared between RSNs. It offers "Presence" scores measuring local overlaps of RSNs for a given ROI (see the WhiteRest tool manual in the Sup. Mat.). It also measures the DiscROver score (for Disconnectome-RSN Overlap score), specifically designed to estimate the white matter disruption of RSNs by a lesion. First, the extent of white matter fibres disconnected by the lesion is estimated using the Disconnectome method49. This method yields a disconnectome map displaying the probability of structural connectivity between the lesion and each brain voxel (Fig. 7a). Hence, the higher the value on the disconnectome map, the more likely the disruption of connectivity in the voxel due to the lesion. Then, the weighted overlap of the RSN (Fig. 7b) with the disconnectome is computed by voxel-wise multiplication of the RSN map and the disconnectome map (Fig. 7c). The DiscROver score is computed as the sum of the values of this weighted overlap map, normalised by the sum of the values in the RSN, and multiplied by 100. With this score, 0 means that the lesion does not impact any white matter voxel of the RSN, and 100 means it impacts the entire RSN.

The complete computation of the DiscROver score is summarised in Eq. 1:

$$DiscROver(RSN, Disco) = 100 \times \frac{\sum_{v \in RSN} Z_{RSN}(v) \times P_{Disco}(v)}{\sum_{v \in RSN} Z_{RSN}(v)} \qquad (1)$$

With "RSN" representing the atlas white matter Z-map of a given RSN, with its voxel values annotated as "$Z_{RSN}(v)$", and "*Disco*" the disconnectome map of a lesion, with its voxel values annotated as "$P_{Disco}(v)$".

**Stroke data analysis**. To validate our WhiteRest atlas, we used the WhiteRest tool to link stroke lesions with RSNs. We first selected 4 RSNs for which we were confident we could identify a specific cognitive function: we chose the somato-motor networks of the right (RSN 08) and left (RSN 09) hand, the language production network (RSN 20), and the language network comprehension (RSN 25). The DiscROver score of the 131 lesions was computed for each of these 4 RSNs.

Each RSN was paired according to their putative function with one of the 3 studied cognitive deficits: The somatomotor network of the right-hand with the right upper-limb motor control deficit; the somatomotor network of the left-hand with the left upper-limb motor control deficit; and the language production and language comprehension networks both with the language deficit.

Because each deficit was associated with multiple clinical scores, we ran a principal component analysis (PCA) on each group of clinical scores and projected the scores on each corresponding first principal component. The "MotorL deficit" score was generated using the 7 clinical scores for left upper-limb motor control deficit. The "MotorR deficit" score was generated using the 7 clinical scores for right upper-limb motor control deficit. And the "Language deficit" score was generated using the 7 clinical scores for language deficit. Additional details on those three sets of scores are available in the supplementary methods. The PCA-derived scores for each cognitive deficit were normalised between 0 and 1, with 0

corresponding to the minimum deficit and 1 to the maximum deficit in the data. This dimensionality reduction step allowed us to capture most of the variance in the data (i.e., the difference in clinical symptoms between patients) while limiting the study of each cognitive deficit to one variable: MotorL, MotorR, and Language deficit scores respectively, explained 95%, 91%, and 74% of the variance of their clinical scores.

Finally, for each RSN-deficit pair, the DiscROver scores of all the patients were plotted against the associated PCA-derived score, with the linear fit and coefficient of determination, uncorrected ($R^2$) and corrected by controlling for age, sex and chronicity ($R^2$-c). Note that the DiscROver scores for the language production network and the language comprehension network were plotted against the same Language deficit score.

**Visualisation**. The 3D z-maps presented in Figs. 1–4 were generated using Surf Ice (https://www.nitrc.org/projects/surfice/), with the default mni152_2009 brain volume as the background template. The 2D brain slices of Figs. 5 and 7 were displayed on a standard template in MRIcron (https://www.nitrc.org/projects/mricron). Each white matter map was masked to remove the grey matter part of the volume and improve readability. The mask used corresponded to voxels defined as white matter in at least 10% of the 150 participants, according to the parcellation provided with the HCP datasets. In Fig. 5, the RSN count was saturated at 14 on the displayed map to improve readability, as only a handful of voxels presented higher values.

**Statistics and reproducibility**. In the stroke analysis, the relationship between neurobehavioral deficits and the RSN DiscROver scores was measured by Pearson's correlation and linear fit. The statistical significance of the correlation was measured using the dedicated function from the Scipy Python library: https://docs.scipy.org/doc/scipy/reference/generated/scipy.stats.pearsonr.html

The confidence intervals (CI) of Fig. 6 represent the 95% CI estimated with 1000 bootstrap resamples of the data, using the "regplot" function from the Seaborn Python library:https://seaborn.pydata.org/generated/seaborn.regplot.html

**Reporting summary**. Further information on research design is available in the Nature Portfolio Reporting Summary linked to this article.

## Data availability
The WhiteRest atlas (non-thresholded maps) is freely available on Neurovault.org, with both the grey matter maps (https://identifiers.org/neurovault.collection:11937) and the white matter maps (https://identifiers.org/neurovault.collection:11895). All fMRI acquisitions are available on the HCP website (https://db.humanconnectome.org/). The data of Figs. 5b and 6 are provided as a Supplementary Data 1.

## Code availability
The WhiteRest module is open-source and freely available as part of the Functionnectome software, which can be found at http://www.bcblab.com or directly downloaded from https://github.com/NotACS/Functionnectome. The MICCA algorithm is also open source and can be freely downloaded from https://www.gin.cnrs.fr/fr/outils/micca/ or directly on the permanent repository https://zenodo.org/record/5837556 (https://doi.org/10.5281/zenodo.5837556).

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

## Acknowledgements

We thank the University of Bordeaux and CNRS for the infrastructural support. This project has received funding from the European Research Council (ERC) under the European Union's Horizon 2020 research and innovation programme (grant agreement No. 818521; MTS), the Marie Skłodowska-Curie grant agreement No. 101028551 (SJF, PERSONALISED) and the Donders Mohrmann Fellowship No. 2401515 (SJF, NEU-ROVARIABILITY). Additionally, this work was conducted in the framework of the University of Bordeaux's IdEx 'Investments for the Future' program RRI 'IMPACT', which received financial support from the French government. Part of the computing was done thanks to the computing facilities MCIA (Mésocentre de Calcul Intensif Aquitain) of the University of Bordeaux and of the Université de Pau et des Pays de l'Adour. Data were provided by the Human Connectome Project, WU-Minn Consortium (Principal Investigators: David Van Essen and Kamil Ugurbil; 1U54MH091657) funded by the 16 NIH Institutes and Centers that support the NIH Blueprint for Neuroscience Research; and by the McDonnell Center for Systems Neuroscience at Washington University.

## Author contributions

V.N. designed the study, implemented the methods, performed the analyses, and wrote the manuscript. S.J.F wrote the manuscript and reviewed the neuroimaging data. L.P. revised the manuscript. L.T. performed analyses and revised the manuscript, M.C. revised the manuscript. M.T.S. co-supervised the study and wrote the manuscript. Marc Joliot co-supervised the study, helped design the study and wrote the manuscript.

## Competing interests

The authors declare no competing interests.

## Additional information

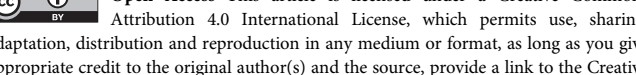

