## [Peer Review File · Communications Biology]

Reviewers' comments:

Reviewer #1 (Remarks to the Author):

This paper by Nozais and colleagues builds on the 'Functionnectome' tool, an open-source python package that generates white matter 4D time series data by projecting the gray matter signal into the white matter. This projection is a weighted sum of the gray matter time series, weighted by the probability of connectivity between white matter and a given gray matter reference voxel ('voxelwise %(dis)connection' from the BCBtoolkit). Here, the authors use Functionnectome to develop an atlas of the white matter tracts supporting resting state networks by performing independent components analysis of the 4D time series data and clustering the components. The authors have made available the white matter and corresponding gray matter atlases that represent 30 resting-state networks (RSNs).

The paper presents an extension of an application that has the potential to be of use to the wider brain mapping community, specifically for investigations into how the white matter supports functional networks at rest, and how these may be disrupted in neurological diseases.

There are a couple of technical faults that reduce the impact of this paper. I have a few suggestions for how these can be addressed:

- Fix the links to the toolbox (they both appear broken - <http://toolkit.bcblab.com/> and <https://storage.googleapis.com/bcblabweb/index.html/BCB/Opendata.html>)
- Include information about how/why only 5 stroke subjects were chosen.
- Justify or provide inclusion criteria for the subset of HCP subjects used for structural connectivity probability maps, as well as more detail on the details of tractography used for healthy control SC data (i.e., probabilistic vs deterministic).
- In their analysis associating stroke symptoms with network disruption, the authors examine the 'RSN identified as primarily affected by the lesion'. It is unclear what is meant by 'primarily affected' in this case – multiple 'presence' scores are available in the WhiteRest tool, e.g. raw, normalized, proportional.
- Include more detailed descriptions of the ICs excluded as artefacts and justification for why a cerebellum IC was not included (include in the methods – I see that this is mentioned in the discussion)
- Information about which MNI model was used should be available in the methods. Reference from: <http://nist.mni.mcgill.ca/atlasses/>

This paper presents a novel method to understand deficits from stroke. However, in my view, the manuscript in its current form does not sufficiently demonstrate the utility of this tool for stroke research, limiting my enthusiasm for the paper.

Currently, the paper does not adequately demonstrate how WhiteRest adds to our understanding of the symptomatology of the stroke patients. The results shown in figure 6 seem to rely on a 'common sense' understanding of post-stroke deficits, in the sense that the dominant RSN identified with WhiteRest ostensibly aligns with the dominant deficit observed in these patients. However, there is no way to understand the variance or magnitude of this relationship given the lack of transparency in how these patients were selected, the extent of their deficits, and the age/sex/chronicity of the subjects.

Additionally, in the case where the lesion appears to overlap with the gray matter, it is not apparent to me how visualizing the white matter network disrupted by the lesion in the occipital cortex (Figure 6e) provides validation for or a more complete understanding of that patient's visuospatial deficits, given

that the lesion may already be expected to impair visuospatial function by disrupting gray matter of the occipital cortex.

Their sample size of 5 subjects also limits my confidence that WhiteRest could be used by a broader audience. Using a larger sample, relating dimensions of symptoms to the extent of disruption to multiple WM networks would demonstrate clinical utility of the tool across multiple functional domains and would provide evidence for their statement that 'lesions to the white matter could severely impact the functioning of multiple RSNs and hence cause a diverse pattern of clinical symptoms.'

Furthermore, for WhiteRest to have immediate relevance to the stroke imaging community, it could be used to link white matter disconnections to the disruption of resting-state functional networks, or some other analysis that demonstrates that the white matter atlas can be used to predict deficits or distant effects of the lesion.

My main critique of the author's conclusions is that they appear to have overlooked prior work characterizing the resting-state BOLD signal in the white matter. The authors claim that a comprehensive description of the white matter circuits in all identifiable resting-state networks is lacking, but I would point them to Huang et al., 2020 (Neuroimage) and Peer et al., 2017 (J Neurosci) who have clustered white matter resting-state fMRI signal in a similar attempt to generate a white matter atlas of resting state networks. These manuscripts directly measure the white matter BOLD signal while the current paper ignores it and instead infers white matter signal based on its connectivity to gray matter. This paper could be improved by highlighting the strengths and uniqueness of their method in the context of this prior literature.

Huang, Y., Yang, Y., Hao, L., Hu, X., Wang, P., Ding, Z., Gao, J.-H., & Gore, J. C. (2020). Detection of functional networks within white matter using independent component analysis. *NeuroImage*, 222, 117278.

Peer, M., Nitzan, M., Bick, A. S., Levin, N., & Arzy, S. (2017). Evidence for Functional Networks within the Human Brain's White Matter. *The Journal of Neuroscience: The Official Journal of the Society for Neuroscience*, 37(27), 6394–6407.

Reviewer #2 (Remarks to the Author):

The authors provided an open-source software to explore the joint contribution of white and grey matter to resting-state networks (RSNs) and facilitate the study of the impact of white matter damage on RSNs. In addition, they presented a small clinical application of the software, to link stroke patients and impacted RSNs, showing that their symptoms aligned well with the estimated functions of the networks.

The data-driven (ICA-based) concurrence of GM and WM RSNs are attractive. I have 3 questions:

1. In the Discussion part, the authors mentioned that "As a proof of concept, the WhiteRest tool was applied to lesions of 5 stroke patients presenting language, motor or attentional deficits. Accordingly, the RSN identified as primarily affected by a lesion aligned with the observed symptoms. Upper limb motor control deficits associated with the RSN of the respective portion of the somato-motor cortex; language deficit associated with a language network for production or comprehension; and visuo-spatial deficits associated with the vision related medial occipital network.". The authors didn't provide

details that how they identify the affection/association. The authors only mentioned "For each patient the RSN with the highest involvement is compared to the clinical outcome 1 year after their stroke" in the Results part. Please provide more details about this procedure.

2. What's the difference between current WhiteRest tool for linking lesion and RSNs and the lesion network mapping (etc., Bowren M, Bruss J, Manzel K, Edwards D, Liu C, Corbetta M, Tranel D, Boes AD. Post-stroke outcomes predicted from multivariate lesion-behaviour and lesion network mapping. *Brain*. 2022 May 24;145(4):1338-1353. doi: 10.1093/brain/awac010. PMID: 35025994)? If they are similar, what's the added value for WhiteRest tool? If they are different, please show us the difference.

3. With regard to the WM RSNs, could the authors tell us, at least using several clear sentences, the difference or similarity between current method and the one introduced by Wang et al. (Wang P, Wang J, Tang Q, Alvarez TL, Wang Z, Kung YC, Lin CP, Chen H, Meng C, Biswal BB. Structural and functional connectivity mapping of the human corpus callosum organization with white-matter functional networks. *Neuroimage*. 2021 Feb 15;227:117642. doi: 10.1016/j.neuroimage.2020.117642. Epub 2020 Dec 15. PMID: 33338619; Wang P, Wang J, Michael A, Wang Z, Klugah-Brown B, Meng C, Biswal BB. White Matter Functional Connectivity in Resting-State fMRI: Robustness, Reliability, and Relationships to Gray Matter. *Cereb Cortex*. 2022 Apr 5;32(8):1547-1559. doi: 10.1093/cercor/bhab181. PMID: 34753176.).

Reviewer #1 (Remarks to the Author):

This paper by Nozais and colleagues builds on the 'Functionnectome' tool, an open-source python package that generates white matter 4D time series data by projecting the gray matter signal into the white matter. This projection is a weighted sum of the gray matter time series, weighted by the probability of connectivity between white matter and a given gray matter reference voxel ('voxelwise %(dis)connection' from the BCBtoolkit). Here, the authors use Functionnectome to develop an atlas of the white matter tracts supporting resting state networks by performing independent components analysis of the 4D time series data and clustering the components. The authors have made available the white matter and corresponding gray matter atlases that represent 30 resting-state networks (RSNs).

The paper presents an extension of an application that has the potential to be of use to the wider brain mapping community, specifically for investigations into how the white matter supports functional networks at rest, and how these may be disrupted in neurological diseases.

1. There are a couple of technical faults that reduce the impact of this paper. I have a few suggestions for how these can be addressed:

- Fix the links to the toolbox (they both appear broken - <http://toolbox.bcblab.com/> and <https://storage.googleapis.com/bcblabweb/index.html/BCB/Opendata.html>)

We thank the reviewer for pointing out this mistake. The website was updated during the review process, breaking those links. The toolbox is now back online at <http://www.bcblab.com>.

2. Include information about how/why only 5 stroke subjects were chosen.

The selection method for the five stroke patients followed a two steps process. First, for specificity, we selected all the patients who showed strong cognitive deficits in one of the four cognitive domains assessed (left motor, right motor, language, visuospatial attention). Then, we computed the Presence score of all the RSN for each patient. We observed a significant correspondence between the symptoms and a high Presence score for RSNs with cognitive functions related to those symptoms. Finally, we selected five representative patients from this group for illustration purposes, who had the highest Presence score.

However, to better answer the reviewer's comments and improve the validation of our atlas, we have decided to replace this analysis with a quantitative one, using the whole stroke dataset (n = 131) available to us (see the response to comment 7). Note that the figures displaying the overlap of RSNs and lesions in patients with strong deficits (originally in Fig. 6) were extended to the 41 corresponding patients (out of 131), and moved to the supplementary material (Supp. Fig. 31-37).

3. Justify or provide inclusion criteria for the subset of HCP subjects used for structural connectivity probability maps, as well as more detail on the details of tractography used for healthy control SC data (i.e., probabilistic vs deterministic).

We now include inclusion criteria for the subset of HCP participants in the methods:

[Page 21 line 494]

The Functionnectome provides default white matter priors³⁰. The white matter priors were originally derived from the 7 Tesla diffusion data of a subset of 100 randomly selected HCP participants from the HCP young adults cohort. Deterministic tractography was run on this diffusion data using StarTrack (<https://www.mr-startrack.com>) to estimate the structural connectivity between each voxel of the brain and build the Functionnectome white matter priors.

4. In their analysis associating stroke symptoms with network disruption, the authors examine the 'RSN identified as primarily affected by the lesion'. It is unclear what is meant by 'primarily affected' in this case – multiple 'presence' scores are available in the WhiteRest tool, e.g. raw, normalized, proportional.

We thank the reviewer for highlighting this imprecision. In the original manuscript, we meant the normalised score (labelled "Presence/RSN (%)” in the output table of the software). However, in our new analysis, we decided to change our approach and use a new score tailored for brain lesion analysis: the DiscROver score. We also now display the score for all the patients (see the response to comment 7).

5. Include more detailed descriptions of the ICs excluded as artefacts and justification for why a cerebellum IC was not included (include in the methods – I see that this is mentioned in the discussion)

We followed the reviewer's suggestion and clarified the methods regarding IC's exclusions:

[Page 22 line 542]

The artefacts were visually identified when the grey matter z-map was spread along the border of the brain mask (typical of motion artefacts, n = 3), or was mainly located in ventricles or along blood vessels (n = 2). The cerebellar RSN was discarded because of known problems with tractography in the cerebellum⁶⁰, to avoid providing the atlas with a white matter map poorly representing the correct connectivity of this region.

6. Information about which MNI model was used should be available in the methods.

Reference from: <http://nist.mni.mcgill.ca/atlasses/>

Thanks, this missing information has been added to the manuscript:

[Page 19 line 453]

... registration to the MNI152 (2009) non-linear asymmetric space. Note that all the analyses done in the present study were conducted in this space, and subsequent mention of "MNI152" will refer to that space.

7. This paper presents a novel method to understand deficits from stroke. However, in my view, the manuscript in its current form does not sufficiently demonstrate the utility of this tool for stroke research, limiting my enthusiasm for the paper.

Currently, the paper does not adequately demonstrate how WhiteRest adds to our understanding of the symptomatology of the stroke patients. The results shown in figure 6 seem to rely on a 'common sense' understanding of post-stroke deficits, in the sense that the dominant RSN identified with WhiteRest ostensibly aligns with the dominant deficit observed in these patients. However, there is no way to understand the variance or magnitude of this relationship given the lack of transparency in how these patients were selected, the extent of their deficits, and the age/sex/chronicity of the subjects.

To address the issues of variance and magnitude of the results, as well as patient selection, we revised our analysis and extended it to all stroke patients (n = 131) rather than a representative subsample. We now provide a quantitative estimation of the patients' neurobehavioral deficits. We also updated the WhiteRest tool to more specifically study the impact of brain lesions. It now includes the DiscROver score (for Disconnectome-RSN Overlap score) to estimate the connectivity disruption of an RSN by a brain lesion.

- Concerning the neurobehavioral deficits, we produced one score per deficit by applying a PCA on the corresponding neurobehavioral assessment scores and then projecting the scores on the associated first principal component. We thus produced the "MotorL deficit" score using the 7 clinical scores related to left upper-limb motor control deficit, the "MotorR deficit" score using the 7 clinical scores related to right upper-limb motor control deficit, and the "Language deficit" score using the 7 clinical scores related to language deficit (see supplementary methods for the names of all 21 scores).
- The new DiscROver score considers the structural connectivity disruption caused by the lesion. First, the lesion mask is used to compute its associated "disconnectome" (Foulon et al., 2018). Then the overlap between the disconnectome and each RSN is used to compute the DiscROver score. The disconnectome of a lesion is a map displaying the white matter fibres disrupted by the lesion based on a normative set of tractography data. The tractography data we used here is the same as the one underlying the white matter priors used to build the WhiteRest atlas. Disconnectome maps better encompass the complete area of white matter affected by a lesion (Talozzi et al., Brain in press).
- Each of the 3 PCA-derived scores was compared to its associated RSN to reveal and measure their relationship.

The results, presented in the new Figure 6, demonstrate a strong relationship between clinical symptoms and related RSN, which we believe reflects the structural-functional accuracy of the WhiteRest atlas and offers validation of the atlas.

We also added information about the sex, age, and chronicity of the patients in the methods.

New method [Page 23 line 564]:

It offers “Presence” scores measuring local overlaps of RSNs for a given ROI (see the WhiteRest tool manual in the Sup. Mat.). It also measures the DiscROver score (for Disconnectome-RSN Overlap score), specifically designed to estimate the white matter disruption of RSNs by a lesion. First, the extent of white matter fibres disconnected by the lesion is estimated using the Disconnectome method⁴⁹. This method yields a disconnectome map displaying the probability of structural connectivity between the lesion and each brain voxel (Fig. 7a). Hence, the higher the value on the disconnectome map, the more likely the disruption of connectivity in the voxel due to the lesion. Then, the weighted overlap of the RSN (Fig. 7b) with the disconnectome is computed by voxel-wise multiplication of the RSN map and the disconnectome map (Fig. 7c). The DiscROver score is computed as the sum of the values of this weighted overlap map, normalised by the sum of the values in the RSN, and multiplied by 100. With this score, 0 means that the lesion does not impact any white matter voxel of the RSN, and 100 means it impacts the entire RSN.

The complete computation of the DiscROver score is summarised in Equation 1:

$$DiscROver(RSN, Disco) = 100 \times \frac{\sum_{v \in RSN} Z_{RSN}(v) \times P_{Disco}(v)}{\sum_{v \in RSN} Z_{RSN}(v)} \quad (1)$$

With “RSN” representing the atlas white matter Z-map of a given RSN, with its voxel values annotated as “ $Z_{RSN}(v)$ ”, and “Disco” the disconnectome map of a lesion, with its voxel values annotated as “ $P_{Disco}(v)$ ”.

Figure 7. Steps for the computation of the DiscROver score. **a** – Lesion mask (left) and associated disconnectome (right). **b** - RSN map used for the DiscROver score computation. **c** - Visual representation of the weighted overlap, and computation of the DiscROver score. Disco: Disconnectome map; RSN: Resting-state network map.

In the results [Page 12 line 247]:

We validated the WhiteRest atlas in a clinical dataset of 131 stroke patients³⁵ and compared their neurobehavioral deficits with the measured impact of the lesion on the RSNs. More specifically, we explored the three deficits which could clearly be associated with RSNs based on their estimated function. The three deficits and the four RSNs in question were: left upper-limb motor control (MotorL) deficit associated with the somatomotor network of the left-hand (RSN 09)(Fig. 6a); right upper-limb motor control (MotorR) deficit associated with the somatomotor network of the right-hand (RSN 08)(Fig. 6b); and language deficit associated with the language comprehension network (RSN 25)(Fig. 6c) and with the language production network (RSN 20)(Fig. 6d). Each deficit score (MotorL, MotorR, and Language deficit) is derived from a principal component analysis (PCA) of the set of neurobehavioral assessment scores related to the deficit. The impact of a lesion on RSNs was measured with the DiscROver score from the WhiteRest tool. We show a strong and highly significant correlation between the neurobehavioral deficit scores and the DiscROver scores for the related RSNs. The Pearson correlation between the scores was: 0.75 ($R^2 = 0.57$) for the “Left upper-limb”; 0.60 ($R^2 = 0.36$) for the “Right upper-limb”; 0.68 ($R^2 = 0.46$) for the “Language (comprehension)”; and 0.61 ($R^2 = 0.37$) for the “Language (production)”. All correlations were highly significant with $p < 10^{-13}$. For a more qualitative overview, we also showed that the lesion of all patients with strong deficits (deficit score in the upper decile, Supp. Fig. 31) overlapped with the studied RSN (Supp. Fig. 32 to 37).

Figure 6: Relationship between neurobehavioral deficit and WhiteRest DiscROver. a-b: Left (a) and right (b) upper-limb motor control deficit vs. DiscROver score for the Somato-motor network of the left (a) and right (b) hand; c-d: Language deficit vs. DiscROver score for the language comprehension network (c) and the language production network (d). In each graph, the blue line corresponds to the linear fit between the scores, and the light blue area corresponds to the confidence interval (set à 95%) for the linear fit. a.u.: arbitrary unit (scores set between 0 and 1); R²: Coefficient of determination.

Additional patients' information in the methods [Page 20 line 465 & 475]:

A dataset of 131 stroke patients (46% female, 54 years +/-11 years, range 19-83 years)...

[The deficits] were established based on the acute (13 ± 4.9 days after stroke) neurobehavioral assessment scores of the patients.

8. Additionally, in the case where the lesion appears to overlap with the gray matter, it is not apparent to me how visualizing the white matter network disrupted by the lesion in the occipital cortex (Figure 6e) provides validation for or a more complete understanding of that patient's visuospatial deficits, given that the lesion may already be expected to impair visuospatial function by disrupting gray matter of the occipital cortex.

We agree with the reviewer that the example of Figure 6e does not accurately reflect the added benefits of studying the impact of the lesion on white matter (rather than grey matter). To properly validate this approach, we repeated all our analyses using the grey matter maps instead of the white matter maps to compute the DiscROver scores. We showed that the correlation between DiscROver scores and clinical scores remained clear (as expected from disconnectome maps in DiscROver computation) but was systematically lower than when using the white matter maps (see **Reply figure 1** below, in comparison with Figure 6 above). Thus, using the white matter maps is both visually compelling (as shown in Supp. Fig. 32-37) and more effective than grey matter maps in terms of quantitative estimation of deficits.

Concerning the visuospatial attention deficit specifically, we have decided to drop it from our analysis as we realised that it may have been slightly overambitious, considering the complexity of this cognitive function and the high number of neurobehavioral scores associated with it (32 scores). To properly model the relationship between clinical assessment and RSN Presence scores would likely require advanced processing tools and procedures to disentangle the relationship between multiple RSNs and visuospatial attention. This would be very interesting to explore but is beyond the scope of the current manuscript.

Reply figure 1: Relationship between neurobehavioral deficit and WhiteRest DiscROver using the grey matter maps. **a-b:** Left (**a**) and right (**b**) upper-limb motor control deficit vs. DiscROver score for the Somato-motor network of the left (**a**) and right (**b**) hand; **c-d:** Language deficit vs. DiscROver score for the language comprehension network (**c**) and for the language production network (**d**). In each graph, the blue line corresponds to the linear fit between the scores, and the light blue area corresponds to the confidence interval (set à 95%) for the linear fit. a.u.: arbitrary unit (scores set between 0 and 1); R^2 : Coefficient of determination.

9. Their sample size of 5 subjects also limits my confidence that WhiteRest could be used by a broader audience. Using a larger sample, relating dimensions of symptoms to the extent of disruption to multiple WM networks would demonstrate clinical utility of the tool across multiple functional domains and would provide evidence for their

statement that ‘lesions to the white matter could severely impact the functioning of multiple RSNs and hence cause a diverse pattern of clinical symptoms.’

As mentioned above, we increased the sample size to 131 patients. This is a sufficient sample size to represent convincing results, as well as more information about the variability between patients.

Concerning our hypothesis of a link between the lesion of “multiple RSNs” and a “diverse pattern of clinical symptoms”, we only have a limited number ($n = 3$) of dimensions of clinical symptoms in our analysis. However, to check the plausibility of our claim, we selected patients for whom at least a third (DiscROver score > 33) of both the right-hand RSN and the language comprehension RSN were impacted. Among these patients ($n = 11$), the majority ($n = 9$) had clear symptoms (i.e., deficit score in the upper quartile) for both language and right upper-limb motor control. While the group size of this analysis is too small for definitive conclusions, we believe these results suggest that our original hypothesis holds some truth and ought to incentivize future research.

[Page 14 line 278]

Using this dataset, we also tested the plausibility of our above-mentioned hypothesis whether lesions impacting multiple RSNs would “cause a diverse pattern of clinical symptoms”. To do so, we selected patients for whom at least a third (DiscROver score > 33) of both the right-hand somatomotor RSN and the language comprehension RSN were impacted. Among these few patients ($n = 11$), the majority ($n = 9$) had clear symptoms (i.e., deficit score in the upper quartile) for both language and right upper-limb motor control (Supp. Fig. 38 & 39). While the group size of this analysis is too small for definitive conclusions, we believe these results suggest that our original hypothesis holds some truth and ought to incentivize future research.

10. Furthermore, for WhiteRest to have immediate relevance to the stroke imaging community, it could be used to link white matter disconnections to the disruption of resting-state functional networks, or some other analysis that demonstrates that the white matter atlas can be used to predict deficits or distant effects of the lesion.

The immediate use of WhiteRest for symptom prediction is, we believe, beyond the scope of the paper. We do envision WhiteRest to be used for this kind of clinical approach, but the current study mainly aims to introduce the atlas. However, as shown in the new results (see the answer to comment 7), the atlas proved to contain relevant and interpretable information linking symptoms with RSNs. This information may be useful in the future to improve existing predictive models and will open the study linking RSN disruption and white matter lesions. However, it will likely require more complex models and a better understanding of the relationship between RSNs and cognition. We added a paragraph in the discussion regarding this matter.

[Page17 line 396]

While promising results link stroke symptoms and RSNs in our study, further investigations will be required to fully disentangle the relationship between cognition (or cognitive deficits) and RSNs, using more advanced models than the relatively simple linear approach from the present study. Recent works have been undertaking the prediction of symptoms and recovery from stroke based on functional and structural data⁵⁸⁻⁵⁹, a very important and interesting goal for which WhiteRest may eventually be of use, adding interpretable data to these multimodal methods.

11. My main critique of the author's conclusions is that they appear to have overlooked prior work characterizing the resting-state BOLD signal in the white matter. The authors claim that a comprehensive description of the white matter circuits in all identifiable resting-state networks is lacking, but I would point them to Huang et al., 2020 (Neuroimage) and Peer et al., 2017 (J Neurosci) who have clustered white matter resting-state fMRI signal in a similar attempt to generate a white matter atlas of resting state networks. These manuscripts directly measure the white matter BOLD signal while the current paper ignores it and instead infers white matter signal based on its connectivity to gray matter. This paper could be improved by highlighting the strengths and uniqueness of their method in the context of this prior literature.

Huang, Y., Yang, Y., Hao, L., Hu, X., Wang, P., Ding, Z., Gao, J.-H., & Gore, J. C. (2020). Detection of functional networks within white matter using independent component analysis. *NeuroImage*, 222, 117278.

Peer, M., Nitzan, M., Bick, A. S., Levin, N., & Arzy, S. (2017). Evidence for Functional Networks within the Human Brain's White Matter. *The Journal of Neuroscience: The Official Journal of the Society for Neuroscience*, 37(27), 6394–6407.

We thank the reviewer for pointing to these very interesting articles. They indeed tackle problems similar to the ones from our study, though in a completely different manner and with different results. We have now appended our paper with these references and discussed the differences, advantages and limitations, between the methods and results.

In the discussion [Page 15 line 316]:

Another intriguing approach to the functional study of white matter has recently been gaining traction and shown auspicious results: the analysis of the BOLD signal directly in the white matter (mini-review by Gore et al., 2019). Using the BOLD signal from white matter allows for its functional exploration and mapping without resorting to connectivity models, which may lead to more physiologically accurate descriptions. Multiple studies have used this framework to unveil RSNs in white matter, successfully adapting classical RSN investigation methods to the white matter³⁹⁻⁴². These studies revealed a functional parcellation of the white matter, showing that it was possible to

identify multiple RSNs purely from functional signals while staying consistent with the underlying structural connectivity. However, these approaches have yet to produce a functional parcellation of the white matter displaying continuous, long-range connectivity between different cortical regions. While efforts have been made to link white matter RSNs with grey matter RSNs, previous studies were unable to present a consistent 1-to-1 correspondence between white and grey matter RSNs. Thus, current analyses using the white matter BOLD signal are limited regarding the functional investigation of the white matter. In contrast, by combining structural and functional (grey matter) signals with the Functionnectome, our approach generated white matter maps that could better represent each network, and systematically paired them with their well-known grey matter counterparts. The WhiteRest atlas also demonstrated overlaps between RSNs, consistent with fibres from distinct networks crossing in the white matter.

References:

30. Nozais, V., Forkel, S. J., Foulon, C., Petit, L. & Thiebaut de Schotten, M. Functionnectome as a framework to analyse the contribution of brain circuits to fMRI. *Commun Biol* 4, 1035 (2021).
 35. Corbetta, M. et al. Common behavioral clusters and subcortical anatomy in stroke. *Neuron* 85, 927–941 (2015).
 39. Peer, M., Nitzan, M., Bick, A. S., Levin, N. & Arzy, S. Evidence for Functional Networks within the Human Brain's White Matter. *J. Neurosci.* 37, 6394–6407 (2017).
 40. Huang, Y. et al. Detection of functional networks within white matter using independent component analysis. *Neuroimage* 222, 117278 (2020).
 41. Wang, P. et al. Structural and functional connectivity mapping of the human corpus callosum organization with white-matter functional networks. *Neuroimage* 227, 117642 (2021).
 42. Wang, P. et al. White Matter Functional Connectivity in Resting-State fMRI: Robustness, Reliability, and Relationships to Gray Matter. *Cereb. Cortex* 32, 1547–1559 (2022).
 49. Foulon, C. et al. Advanced lesion symptom mapping analyses and implementation as BCBtoolkit. *Gigascience* 7, 1–17 (2018).
 58. Talozzi, L. et al. Latent disconnectome prediction of long-term cognitive symptoms in stroke. *Brain* (in press).
 59. Bowren, M. et al. Post-stroke outcomes predicted from multivariate lesion-behaviour and lesion network mapping. *Brain* 145, 1338–1353 (2022).
 60. Catani, M. From hodology to function. *Brain: a journal of neurology* vol. 130 602–605 (2007).
- Talozzi, L. et al., Latent disconnectome prediction of long-term cognitive symptoms in stroke. *Brain* (in press).

Reviewer #2 (Remarks to the Author):

The authors provided an open-source software to explore the joint contribution of white and grey matter to resting-state networks (RSNs) and facilitate the study of the impact of white matter damage on RSNs. In addition, they presented a small clinical application of the software, to link stroke patients and impacted RSNs, showing that their symptoms aligned well with the estimated functions of the networks.

The data-driven (ICA-based) concurrence of GM and WM RSNs are attractive. I have 3 questions:

1. In the Discussion part, the authors mentioned that “As a proof of concept, the WhiteRest tool was applied to lesions of 5 stroke patients presenting language, motor or attentional deficits. Accordingly, the RSN identified as primarily affected by a lesion aligned with the observed symptoms. Upper limb motor control deficits associated with the RSN of the respective portion of the somato-motor cortex; language deficit associated with a language network for production or comprehension; and visuo-spatial deficits associated with the vision related medial occipital network.”. The authors didn’t provide details that how they identify the affection/association. The authors only mentioned “For each patient the RSN with the highest involvement is compared to the clinical outcome 1 year after their stroke” in the Results part. Please provide more details about this procedure.

We thank the reviewer for highlighting the lack of precision on that part. The original selection method for the 5 stroke patients followed a two steps process. First, we selected all the patients who showed a salient cognitive deficit in one of four cognitive domains (i.e. left motor, right motor, language, visuospatial attention), defined by clinical scores: the different neurobehavioral assessment scores were examined, and the scores displaying values significantly different from the norm were considered symptomatic of stroke. Then, we computed the Presence score of all the RSN for each patient. We observed a significant correspondence between the symptoms and a high Presence score for RSNs with cognitive function related to those symptoms. Finally, we selected 5 patients from this group for illustration purposes.

However, in the revised manuscript, we decided to change our approach to better explore the relationship between RSN, white matter lesions, and clinical symptoms. The new procedure now presents quantitative results for all the patients in our clinical dataset (n = 131), with a properly detailed method section. In particular, we now use the DiscROver score to estimate the impact of a lesion on RSNs. This new score was designed specifically to measure lesion-related structural disconnection by taking whole-brain structural connectivity into account, as opposed to the purely local Presence score. Note that the figures displaying the overlap of RSNs and lesions in patients with strong deficits (originally in Fig. 6) were extended to the 41 corresponding patients (out of 131), and moved to the supplementary material (Supp. Fig. 31-37).

New method [Page 24 line 591]:

To validate our WhiteRest atlas, we used the WhiteRest tool to link stroke lesions with RSNs. We first selected 4 RSNs for which we were confident we could identify a specific cognitive function: we chose the somatomotor networks of the right (RSN 08) and left (RSN 09) hand, the language production network (RSN 20), and the language network comprehension (RSN 25). The DiscROver score of the 131 lesions was computed for each of these 4 RSNs.

Each RSN was paired according to their putative function with one of the 3 studied cognitive deficits: The somatomotor network of the right-hand with the right upper-limb motor control deficit; the somatomotor network of the left-hand with the left upper-limb motor control deficit; and the language production and language comprehension networks both with the language deficit.

Because each deficit was associated with multiple clinical scores, we ran a principal component analysis (PCA) on each group of clinical scores and projected the scores on each corresponding first principal component. The “MotorL deficit” score was generated using the 7 clinical scores for left upper-limb motor control deficit. The “MotorR deficit” score was generated using the 7 clinical scores for right upper-limb motor control deficit. And the “Language deficit” score was generated using the 7 clinical scores for language deficit. Additional details on those three sets of scores are available in the supplementary methods. The PCA-derived scores for each cognitive deficit were normalised between 0 and 1, with 0 corresponding to the minimum deficit and 1 to the maximum deficit in the data. This dimensionality reduction step allowed us to capture most of the variance in the data (i.e., the difference in clinical symptoms between patients) while limiting the study of each cognitive deficit to one variable: MotorL, MotorR, and Language deficit scores respectively, explained 95%, 91%, and 74% of the variance of their clinical scores.

Finally, for each RSN-deficit pair, the DiscROver scores of all the patients were plotted against the associated PCA-derived score, with the linear fit and coefficient of determination (R^2). Note that the DiscROver scores for the language production network and the language comprehension network were plotted against the same Language deficit score.

2. What’s the difference between current WhiteRest tool for linking lesion and RSNs and the lesion network mapping (etc., Bowren M, Bruss J, Manzel K, Edwards D, Liu C, Corbetta M, Tranel D, Boes AD. Post-stroke outcomes predicted from multivariate lesion-behaviour and lesion network mapping. *Brain*. 2022 May 24;145(4):1338-1353. doi: 10.1093/brain/awac010. PMID: 35025994)? If they are similar, what’s the added value for WhiteRest tool? If they are different, please show us the difference.

We thank the reviewer for bringing to our attention this very interesting paper. While at first there may appear to be some similarities between the two studies, both the methods and the investigated questions are fundamentally different. In their work,

Bowren and colleagues used structural and functional connectivity data to define structural and functional networks specific to certain deficits, a lesion-network mapping (LNM) approach. In contrast, the WhiteRest tool uses the structural-functional networks of the WhiteRest atlas. While not as specific to lesion study as Bowren and colleagues', the WhiteRest approach offers the advantage of being more easily interpretable (in terms of impacted network). It makes it easier to link lesion and deficit to the existing resting-state network literature. In the end, the two tools have many different goals and should not be competing against each other: the LNM approach was designed for the prediction of clinical outcomes, while our RSN atlas approach was designed to improve our understanding of the triple relationship between RSNs, cognition, and lesion symptoms.

We added this reference to our discussion to clarify that point.

[Page17 line 396]

While promising results link stroke symptoms and RSNs in our study, further investigations will be required to fully disentangle the relationship between cognition (or cognitive deficits) and RSNs, using more advanced models than the relatively simple linear approach from the present study. Recent works have been undertaking the prediction of symptoms and recovery from stroke based on functional and structural data⁵⁸⁻⁵⁹, a very important and interesting goal for which WhiteRest may eventually be of use, adding interpretable data to these multimodal methods.

3. With regard to the WM RSNs, could the authors tell us, at least using several clear sentences, the difference or similarity between current method and the one introduced by Wang et al. (Wang P, Wang J, Tang Q, Alvarez TL, Wang Z, Kung YC, Lin CP, Chen H, Meng C, Biswal BB. Structural and functional connectivity mapping of the human corpus callosum organization with white-matter functional networks. *Neuroimage*. 2021 Feb 15;227:117642. doi: 10.1016/j.neuroimage.2020.117642. Epub 2020 Dec 15. PMID: 33338619; Wang P, Wang J, Michael A, Wang Z, Klugah-Brown B, Meng C, Biswal BB. White Matter Functional Connectivity in Resting-State fMRI: Robustness, Reliability, and Relationships to Gray Matter. *Cereb Cortex*. 2022 Apr 5;32(8):1547-1559. doi: 10.1093/cercor/bhab181. PMID: 34753176.).

We thank the reviewer for suggesting these articles. The works from Wang and colleagues focuses on the direct analysis of BOLD signal in white matter (i.e. a 'measured' white matter BOLD signal), whereas we are using the Functionnectome approach in which we project grey matter BOLD signal onto the white matter (i.e.a 'modelled' white matter BOLD signal). Exploring measured white matter BOLD signals is a relatively new and exciting field of research, and, as shown in the mentioned articles, it was successfully related to RSNs. However, this kind of signal is still quite faint and noisy (compared to grey matter BOLD), with some remaining doubts concerning its biological origin. By itself, it has yet to produce a functional parcellation of white matter displaying continuous, long-range connectivity between different cortical regions (as can be seen in the RSNs proposed by the 2021 paper from Wang

and colleagues). Such results are still very interesting, but will probably not be enough, for instance, to measure the impact of a white matter lesion on RSNs and cognition. By combining structural and stronger functional (grey matter) signals, the Functionnectome approach we use allowed us to generate more extensive white matter maps, which better represent the full extent of each network. This was, of course, at the cost of using a modelling approach of the connectivity (the tractography and the Functionnectome algorithms) instead of the raw white matter BOLD signal. An interesting idea to bridge the gap between these two approaches would be to use our Functionnectome method to predict the BOLD signal in white matter voxels, and use this predicted, modelled, signal to confirm, or maybe denoise, the measured white matter BOLD signal.

The other main advantage of our approach is that it allows for the pairing of the familiar grey matter RSNs with our new white matter RSNs, vastly facilitating their interpretation and integration in a cognitive-focused framework (i.e., linking white matter to RSNs, and RSNs to cognition). The 2022 study by Wang and colleagues offers some relationship between grey and white matter RSNs, but not a 1-to-1 association as we are doing in the present atlas.

We agree with the reviewer that this point should have been brought up in the study, so we now discuss it in the revised manuscript.

In the discussion [Page 15 line 316]:

Another intriguing approach to the functional study of white matter has recently been gaining traction and shown auspicious results: the analysis of the BOLD signal directly in the white matter (mini-review by Gore et al., 2019). Using the BOLD signal from white matter allows for its functional exploration and mapping without resorting to connectivity models, which may lead to more physiologically accurate descriptions. Multiple studies have used this framework to unveil RSNs in white matter, successfully adapting classical RSN investigation methods to the white matter³⁹⁻⁴². These studies revealed a functional parcellation of the white matter, showing that it was possible to identify multiple RSNs purely from functional signals while staying consistent with the underlying structural connectivity. However, these approaches have yet to produce a functional parcellation of the white matter displaying continuous, long-range connectivity between different cortical regions. While efforts have been made to link white matter RSNs with grey matter RSNs, previous studies were unable to present a consistent 1-to-1 correspondence between white and grey matter RSNs. Thus, current analyses using the white matter BOLD signal are limited regarding the functional investigation of the white matter. In contrast, by combining structural and functional (grey matter) signals with the Functionnectome, our approach generated white matter maps that could better represent each network, and systematically paired them with their well-known grey matter counterparts. The WhiteRest atlas also demonstrated overlaps between RSNs, consistent with fibres from distinct networks crossing in the white matter.

References:

39. Peer, M., Nitzan, M., Bick, A. S., Levin, N. & Arzy, S. Evidence for Functional Networks within the Human Brain's White Matter. *J. Neurosci.* 37, 6394–6407 (2017).
40. Huang, Y. et al. Detection of functional networks within white matter using independent component analysis. *Neuroimage* 222, 117278 (2020).
41. Wang, P. et al. Structural and functional connectivity mapping of the human corpus callosum organization with white-matter functional networks. *Neuroimage* 227, 117642 (2021).
42. Wang, P. et al. White Matter Functional Connectivity in Resting-State fMRI: Robustness, Reliability, and Relationships to Gray Matter. *Cereb. Cortex* 32, 1547–1559 (2022).
58. Talozzi, L. et al. Latent disconnectome prediction of long-term cognitive symptoms in stroke. *Brain* (in press).
59. Bowren, M. et al. Post-stroke outcomes predicted from multivariate lesion-behaviour and lesion network mapping. *Brain* 145, 1338–1353 (2022).

Reviewers' comments:

Reviewer #1 (Remarks to the Author):

The paper is greatly improved and I commend the authors' success in addressing all points of feedback from both reviewers. I have just one point about the DiscROver analysis that was added to the paper.

One of the main findings of the paper is that most of the brain's white matter is shared by multiple resting-state networks. This result in fact undermines their applied results, which argue that neurobehavioral deficits are associated with damage to specific white matter networks. To provide evidence that deficits are linked to specific white matter network damage, the authors might show associations between the DiscROver score of all RSNs (not just a priori selected networks) and motor/language deficits. If there is a strong correlation between a deficit and several, functionally unrelated WM RSNs, one might wonder whether the deficit-DiscROver associations are driven by real structure-function relationships or simply by lesion volume.

Reviewer #2 (Remarks to the Author):

Thanks for the response of the authors. The authors partially answered my questions. I still have doubts that cannot be resolved by the answers.

1. The authors mentioned that "The new procedure now presents quantitative results for all the patients in our clinical dataset ($n = 131$), with a properly detailed method section. In particular, we now use the DiscROver score to estimate the impact of a lesion on RSNs. This new score was designed specifically to measure lesion-related structural disconnection by taking whole-brain structural connectivity into account, as opposed to the purely local Presence score. Note that the figures displaying the overlap of RSNs and lesions in patients with strong deficits (originally in Fig. 6) were extended to the 41 corresponding patients (out of 131), and moved to the supplementary material (Supp. Fig. 31-37)". Only presenting the spatial maps of the patients showing strong deficits is not enough. Since the authors declaimed that lesions impacting multiple RSNs would "cause a diverse pattern of clinical symptoms", why not also present the overlap of RSNs and lesions in patients with mild deficits, in comparison with the strong deficits ones, to highlight the statement?

In the revised manuscript, the authors mentioned that "Because each deficit was associated with multiple clinical scores, we ran a principal component analysis (PCA) on each group of clinical scores and projected the scores on each corresponding first principal component". (line 531-533). Why only choose the first principal component, not the first two or first three? What's the criteria of the authors for the selection? Variance keeping or scree plot?

2. Question 2 solved.

3. In the revised manuscript, line 288-289, the authors said that "While efforts have been made to link white matter RSNs with grey matter RSNs, previous studies were unable to present a consistent 1-to-1 correspondence between white and grey matter RSNs.". Indeed, we don't know whether the correspondence between white and grey matter RSNs is 1-to-1 or 1-to-many or many-to-1, why the authors think that 1-to-1 relationship is better than other possible relationships and thus could demonstrate the advantage of current model?

Beyond these 3 questions, while reading the revised manuscript, there are several new questions in my mind:

4. For stroke dataset, the authors only provide the age, sex and chronicity without using them to get the adjust R2. This cannot convince the readers that the adjust R2 (the one we usually want to know) is really good or not. Please add this information.

5. For Figure 7, if I understand correctly, c is the overlapping of a and b. Why for c there is region (top left, close to the ventricle) that should exist? Please correct me if my understanding is not right.

6. The authors mentioned that "To do so, we selected patients for whom at least a third (DiscROver score > 33) of both the right-hand somatomotor RSN and the language comprehension RSN were impacted. Among these few patients (n = 11), the majority (n = 9) had clear symptoms (i.e., deficit score in the upper quartile) for both language and right upper-limb motor control (Supp. Fig. 38 & 39)." If the model only work for the extreme situation, like the patients with severe symptoms but no evidence for other patients with mild symptoms, we cannot say "these preliminary results are a strong indication that our original hypothesis holds some truth" and the added value for clinic is still not clear.

7. There is a typo-error:

Line 409, "A dataset of 131 stroke patients (46% female, 54 years +/-11 years, range 19-83 years)" should be corrected as "A dataset of 131 stroke patients (46% female, 54 years \pm 11 years, range 19-83 years)"

Reviewer #1

The paper is greatly improved and I commend the authors' success in addressing all points of feedback from both reviewers. I have just one point about the DiscROver analysis that was added to the paper.

One of the main findings of the paper is that most of the brain's white matter is shared by multiple resting-state networks. This result in fact undermines their applied results, which argue that neurobehavioral deficits are associated with damage to specific white matter networks. To provide evidence that deficits are linked to specific white matter network damage, the authors might show associations between the DiscROver score of all RSNs (not just a priori selected networks) and motor/language deficits. If there is a strong correlation between a deficit and several, functionally unrelated WM RSNs, one might wonder whether the deficit-DiscROver associations are driven by real structure-function relationships or simply by lesion volume.

We appreciate the reviewer's insightful comment regarding the correlation between a deficit and the DiscROver scores of functionally unrelated RSNs. However, due to the limited understanding of the cognitive roles of each RSN, it is difficult to determine which RSNs are functionally unrelated. Nonetheless, our WhiteRest tool can link RSNs with clinical observations, allowing for a circular approach using the traditional clinico-anatomical framework. By using clinical observations to better understand the function of RSNs, we can improve our understanding of the impact of lesions on cognition through damaged RSNs.

To ensure thoroughness, we also checked the effect of lesion volume on DiscROver scores. While we expected lesion's volume to be a factor in the score, our findings showed that most of the explained variance did not come from the lesion's volume. This result supports the notion that the DiscROver score contains specific information important for predicting symptoms. We compared the R² between the deficits and DiscROver scores, as shown in the manuscript, with the R² after correction by lesion volume to obtain these results.

	R ²	R ² corrected
Right hand	0.36	0.35
Left hand	0.57	0.50
Language comprehension	0.46	0.36
Language production	0.37	0.27

Reviewer #2 (Remarks to the Author):

Thanks for the response of the authors. The authors partially answered my questions. I still have doubts that cannot be resolved by the answers.

1. The authors mentioned that “The new procedure now presents quantitative results for all the patients in our clinical dataset (n = 131), with a properly detailed method section. In particular, we now use the DiscROver score to estimate the impact of a lesion on RSNs. This new score was designed specifically to measure lesion-related structural disconnection by taking whole-brain structural connectivity into account, as opposed to the purely local Presence score. Note that the figures displaying the overlap of RSNs and lesions in patients with strong deficits (originally in Fig. 6) were extended to the 41 corresponding patients (out of 131), and moved to the supplementary material (Supp. Fig. 31-37).”. Only presenting the spatial maps of the patients showing strong deficits is not enough. Since the authors declaimed that lesions impacting multiple RSNs would “cause a diverse pattern of clinical symptoms”, why not also present the overlap of RSNs and lesions in patients with mild deficits, in comparison with the strong deficits ones, to highlight the statement?

Mild deficits tend to be linked with lower DiscROver scores, as shown by the linear relationship between deficits and DiscROver scores. In practice, lesion/RSN overlap for mild deficits mostly shows that low deficits are linked with low overlaps. To illustrate this point, for each deficit type, we added to the supplementary figures the overlap figures of 10 randomly selected patients among the patients displaying a mild cognitive impairment.

In the revised manuscript, the authors mentioned that “Because each deficit was associated with multiple clinical scores, we ran a principal component analysis (PCA) on each group of clinical scores and projected the scores on each corresponding first principal component”. (line 531-533). Why only choose the first principal component, not the first two or first three? What’s the criteria of the authors for the selection? Variance keeping or scree plot?

We agree that keeping more than just the first component can be useful in some cases, but in our analysis the overwhelming majority of the variance was explained by the first component, as mentioned in the methods line 614-615 “MotorL, MotorR, and Language deficit scores respectively, explained 95%, 91%, and 74% of the variance”. In this context, we decided that keeping more than the first components was unnecessary.

For more details, here are the explained variance for the three first components in each analysis:

Explained variance	PC1	PC2	PC3
motorL	95%	2.6%	2.3%
motorR	91%	4.7%	3.4%
lang	74%	13%	7.8%

We acknowledge that this method may result in some loss of information from the original clinical scores, particularly when it comes to language deficits. As a result, we plan to conduct further research to investigate the connection between each RSN's DiscROver score and all neurobehavioral assessment scores.

2. Question 2 solved.

3. In the revised manuscript, line 288-289, the authors said that “While efforts have been made to link white matter RSNs with grey matter RSNs, previous studies were unable to present a consistent 1-to-1 correspondence between white and grey matter RSNs.”. Indeed, we don't know whether the correspondence between white and grey matter RSNs is 1-to-1 or 1-to-many or many-to-1, why the authors think that 1-to-1 relationship is better than other possible relationships and thus could demonstrate the advantage of current model?

That's an interesting point. Our assumption is that WM RSN are directly related to the GM RSN, as they would reflect the WM connectivity supporting classical RSNs. We believe this is the most parsimonious explanation for WM RSN, but we do not exclude other organizations in the WM (although we haven't found an alternative theoretical framework that would explain 1-to-many or many-to-1 relationships). To better encompass all possibilities, we rephrased and nuanced the related part of the discussion.

In the discussion:

“

As our objective is to investigate the white matter connectivity underlying traditional grey matter RSNs, the analyses directly using white matter BOLD signals do not appear to offer immediately interpretable results on this matter. In contrast, by combining structural and functional (grey matter) signals with the Functionnectome, our approach generated white matter maps that could better represent each network, and systematically paired them with their well-known grey matter counterparts. The WhiteRest atlas also demonstrated overlaps between RSNs, consistent with fibres from distinct networks crossing in the white matter. Nevertheless, combining both approaches in the future (white matter BOLD analysis and Functionnectome) could be highly beneficial as it could allow for a finer understanding of the functional involvement of white matter in resting-state.

“

Beyond these 3 questions, while reading the revised manuscript, there are several new questions in my mind:

4. For stroke dataset, the authors only provide the age, sex and chronicity without using them to get the adjust R2. This cannot convince the readers that the adjust R2 (the one we usually want to know) is really good or not. Please add this information.

Indeed, this analysis was missing. We amended the methods and added the corrected R2 to the figures. Note, however, that the impact of the correction was minimal and did not change our conclusions.

Updated R2:

Deficit	RSN	R ²	R ² -corrected.
Motor Left	RSN09	0.57	0.54
Motor Right	RSN08	0.36	0.36
Language comprehension	RSN25	0.46	0.44
Language production	RSN20	0.37	0.36

In the methods:

“

Finally, for each RSN-deficit pair, the DiscROver scores of all the patients were plotted against the associated PCA-derived score, with the linear fit and coefficient of determination, **uncorrected (R²) and corrected by controlling for age, sex and chronicity (R²-corrected)**.

“

5. For Figure 7, if I understand correctly, c is the overlapping of a and b. Why for c there is region (top left, close to the ventricle) that should exist? Please correct me if my understanding is not right.

Well spotted. This was due to an error in the figure. We have corrected it and updated the figure.

6. The authors mentioned that “To do so, we selected patients for whom at least a third (DiscROver score > 33) of both the right-hand somatomotor RSN and the language comprehension RSN were impacted. Among these few patients (n = 11), the majority (n = 9) had clear symptoms (i.e., deficit score in the upper quartile) for both language and right upper-limb motor control (Supp. Fig. 38 & 39).”. If the model only work for the extreme situation, like the patients with severe symptoms but no evidence for other patients with mild symptoms, we cannot say “these preliminary results are a strong indication that our original hypothesis holds some truth” and the added value for clinic is still not clear.

While we agree that studying patients with mild deficits is of great importance for clinical applications of any tool, the patients with mild deficits in our dataset were difficult to identify, as the limit between “mild deficits” and “no significant deficits” is quite hard to define (especially in the acute phase). We chose the upper quartile of the deficit scores as it allowed us to select patients who showed what we believe to be “clear” symptoms.

But to better reflect the current state of the clinical potential of WhiteRest, we rephrased this part of the discussion to add more nuance to our statement.

At the end of the results:

“

While the group size of this analysis is too small for definitive conclusions and limited to two RSNs, we believe these preliminary results are encouraging and ought to incentivize more research into this issue. It should however be noted that they are not enough to validate direct clinical applications of the atlas, but do show promise for its potential use as a tool in clinical research.

“

7. There is a typo-error:

Line 409, “A dataset of 131 stroke patients (46% female, 54 years +/-11 years, range 19-83 years)” should be corrected as “A dataset of 131 stroke patients (46% female, 54 years \pm 11 years, range 19-83 years)”

This was corrected.

New figures supp mat

Supplementary Figure 39: Ten patients with mild left upper-limb motor control deficit. **a-j:** In red, the stroke lesion; in green, the somato-motor network for the left hand (RSN 09).

Supplementary Figure 40: Ten patients with mild right upper-limb motor control deficit. **a-j:** In red, the stroke lesion; in green, the somato-motor network for the left hand (RSN 08).

Supplementary Figure 41: Patients with mild language deficit (part 1/2). Left column (**a-c-e-g-i**): Display with the Language prediction network (RSN 20). Right column (**b-d-f-h-j**): Display with the Language comprehension network (RSN 25). Each line represents the same patient. In red, the stroke lesion; in green, the RSN.

Supplementary Figure 42: Patients with mild language deficit (part 2/2). Left column (**a-c-e-g-i**): Display with the Language prediction network (RSN 20). Right column (**b-d-f-h-j**): Display with the Language comprehension network (RSN 25). Each line represents the same patient. In red, the stroke lesion; in green, the RSN.

REVIEWERS' COMMENTS:

Reviewer #2 (Remarks to the Author):

The paper is much improved and my comments have been successfully addressed.